# Exploring a Principled Framework for Deep Subspace Clustering

**Xianghan Meng[†], Zhiyuan Huang[†] & Wei He**
Beijing University of Posts and Telecommunications, Beijing 100876, P.R. China
{mengxianghan,huangzhiyuan,wei.he}@bupt.edu.cn

**Xianbiao Qi & Rong Xiao**
Intellifusion, Shenzhen, P.R. China

**Chun-Guang Li[*]**
Beijing University of Posts and Telecommunications, Beijing 100876, P.R. China
lichunguang@bupt.edu.cn

## Abstract

Subspace clustering is a classical unsupervised learning task, built on a basic assumption that high-dimensional data can be approximated by a union of subspaces (UoS). Nevertheless, the real-world data are often deviating from the UoS assumption. To address this challenge, state-of-the-art deep subspace clustering algorithms attempt to jointly learn UoS representations and self-expressive coefficients. However, the general framework of the existing algorithms suffers from a catastrophic feature collapse and lacks a theoretical guarantee to learn desired UoS representation. In this paper, we present a Principled fRamewOrk for Deep Subspace Clustering (PRO-DSC), which is designed to learn structured representations and self-expressive coefficients in a unified manner. Specifically, in PRO-DSC, we incorporate an effective regularization on the learned representations into the self-expressive model, prove that the regularized self-expressive model is able to prevent feature space collapse, and demonstrate that the learned optimal representations under certain condition lie on a union of orthogonal subspaces. Moreover, we provide a scalable and efficient approach to implement our PRO-DSC and conduct extensive experiments to verify our theoretical findings and demonstrate the superior performance of our proposed deep subspace clustering approach.

## 1 Introduction

Subspace clustering is an unsupervised learning task, aiming to partition high dimensional data that are approximately lying on a union of subspaces (UoS), and finds wide-ranging applications, such as motion segmentation (Costeira & Kanade, 1998; Vidal et al., 2008; Rao et al., 2010), hybrid system identification (Vidal, 2004; Bako & Vidal, 2008), image representation and clustering (Hong et al., 2006; Lu et al., 2012), genes expression clustering (McWilliams & Montana, 2014) and so on.

Existing subspace clustering algorithms can be roughly divided into four categories: iterative methods (Tseng, 2000; Ho et al., 2003; Zhang et al., 2009), algebraic geometry based methods (Vidal et al., 2005; Tsakiris & Vidal, 2017), statistical methods (Fischler & Bolles, 1981), and spectral clustering-based methods (Chen & Lerman, 2009; Elhamifar & Vidal, 2009; Liu et al., 2010; Lu et al., 2012; You et al., 2016a; Zhang et al., 2021). Among them, spectral clustering based methods gain the most popularity due to the broad theoretical guarantee and superior performance.

The vital component in spectral clustering based methods is a so-called *self-expressive* model (Elhamifar & Vidal, 2009; 2013). Formally, given a dataset $\mathcal{X} := \{\boldsymbol{x}_1, \cdots, \boldsymbol{x}_N\}$ where $\boldsymbol{x}_j \in \mathbb{R}^D$,

---

[*]Corresponding author. [†] These two authors are equally contributed.

self-expressive model expresses each data point $\boldsymbol{x}_j$ by a linear combination of other points, i.e.,

$$\boldsymbol{x}_j = \sum_{i \neq j} c_{ij} \boldsymbol{x}_i, \tag{1}$$

where $c_{ij}$ is the corresponding self-expressive coefficient. The most intriguing merit of the self-expressive model is that the solution of the self-expressive model under proper regularizer on the coefficients $c_{ij}$ is guaranteed to satisfy a subspace-preserving property, namely, $c_{ij} \neq 0$ *only if* $\boldsymbol{x}_i$ *and* $\boldsymbol{x}_j$ *are in the same subspace* (Elhamifar & Vidal, 2013; Soltanolkotabi & Candes, 2012; Li et al., 2018). Having had the optimal self-expressive coefficients $\{c_{ij}\}_{i,j=1}^N$, the data affinity can be induced by $|c_{ij}| + |c_{ji}|$ for which spectral clustering is applied to yield the partition of the data.

Despite the broad theoretical guarantee, the vanilla self-expressive model still faces great challenges when applied to the complex real-world data that may not well align with the UoS assumption. Earlier works devote to address this deficiency by learning a linear transform of the data (Patel et al., 2013; 2015) or introducing a nonlinear kernel mapping (Patel & Vidal, 2014) under which the representations of the data are supposed to be aligned with the UoS assumption. However, there is a lack of principled mechanism to guide the learning of the linear transforms or the design of the nonlinear kernels to guarantee the representations of the data to form a UoS structure.

To handle complex real-world data, in the past few years, there is a surge of interests in designing deep subspace clustering frameworks, e.g., (Ji et al., 2017; Peng et al., 2018; Zhou et al., 2018; Zhang et al., 2019a; Dang et al., 2020; Peng et al., 2020; Lv et al., 2021; Wang et al., 2023b; Zhao et al., 2024). In these works, usually a deep neural network-based representation learning module is integrated to the self-expressive model, to learn the representations $\boldsymbol{Z} \in \mathbb{R}^{d \times N}$ and the self-expressive coefficients $\boldsymbol{C} = \{c_{ij}\}_{i,j=1}^N$ in a joint optimization framework. However, as analyzed in (Haeffele et al., 2021) that, the optimal representations $\boldsymbol{Z}$ of these methods tend to catastrophically collapse into subspaces with dimensions much lower than the ambient space, which is detrimental to subspace clustering and there is no evidence that the learned representations form a UoS structure.

In this paper, we attempt to propose a Principled fRamewOrk for Deep Subspace Clustering (PRO-DSC), which is able to simultaneously learn structured representations and self-expressive coefficients. Specifically, in PRO-DSC, we incorporate an effective regularization on the learned representations into the self-expressive model and prove that our PRO-DSC can effectively prevent feature collapse. Moreover, we demonstrate that our PRO-DSC under certain condition can yield structured representations forming a UoS structure and provide a scalable and efficient approach to implement PRO-DSC. We conduct extensive experiments on the synthetic data and six benchmark datasets to verify our theoretical findings and the superior performance of our proposed approach.

**Contributions.** The contributions of the paper are highlighted as follows.

1. We propose a Principled fRamewOrk for Deep Subspace Clustering (PRO-DSC) that learns both structured representations and self-expressive coefficients simultaneously, in which an effective regularization on the learned representations is incorporated to prevent feature space collapse.

2. We provide a rigorous analysis for the optimal solution of our PRO-DSC, derive a sufficient condition that guarantees the learned representations to escape from feature collapse, and further demonstrate that our PRO-DSC under certain condition can yield structured representations of a UoS structure.

3. We conduct extensive experiments to verify our theoretical findings and to demonstrate the superior performance of the proposed approach.

To the best of our knowledge, this is the first principled framework for deep subspace clustering that is guaranteed to prevent feature collapse problem and is shown to yield the UoS representations.

## 2 DEEP SUBSPACE CLUSTERING: A PRINCIPLED FRAMEWORK, JUSTIFICATION, AND IMPLEMENTATION

In this section, we review the popular framework for deep subspace clustering, called Self-Expressive Deep Subspace Clustering (SEDSC) at first, then present our principled framework for deep subspace clustering and provide a rigorous characterization of the optimal solution and the

property of the learned structured representations. Finally we describe a scalable implementation based on differential programming for the proposed framework. Please refer to Appendix A for the detailed proofs of our theoretical results.

## 2.1 Prerequisite

To apply subspace clustering to complex real-world data that may not well align with the UoS assumption, there has been a surge of interests in exploiting deep neural networks to learn representations and then apply self-expressive model to the learned representations, e.g., (Peng et al., 2016; 2018; Ji et al., 2017; Zhou et al., 2018; Zhang et al., 2019a; Dang et al., 2020; Peng et al., 2020; Lv et al., 2021; Wang et al., 2023b; Zhao et al., 2024).

Formally, the optimization problem of these SEDSC models can be formulated as follows:[1]

$$\min_{\boldsymbol{Z},\boldsymbol{C}} \quad \frac{1}{2}\|\boldsymbol{Z} - \boldsymbol{Z}\boldsymbol{C}\|_F^2 + \beta \cdot r(\boldsymbol{C}) \quad \text{s.t.} \quad \|\boldsymbol{Z}\|_F^2 = N, \tag{2}$$

where $\boldsymbol{Z} \in \mathbb{R}^{d \times N}$ denotes the learned representation, $\boldsymbol{C} \in \mathbb{R}^{N \times N}$ denotes the self-expressive coefficient matrix, and $\beta > 0$ is a hyper-parameter. The following lemma characterizes the property of the optimal solution $\boldsymbol{Z}$ for problem (2).

**Lemma 1** (Haeffele et al., 2021). *The rows of the optimal solution $\boldsymbol{Z}$ for problem (2) are the eigenvectors that associate with the smallest eigenvalues of $(\boldsymbol{I} - \boldsymbol{C})(\boldsymbol{I} - \boldsymbol{C})^\top$.*

In other words, the optimal representation $\boldsymbol{Z}$ in SEDSC is restricted to an extremely "narrow" subspace whose dimension is much smaller than $d$, leading to an undesirable collapsed solution. [2]

## 2.2 Our Principled Framework for Deep Subspace Clustering

In this paper, we attempt to propose a principled framework for deep subspace clustering that provably learns structured representations with maximal intrinsic dimensions.

To be specific, we try to optimize the self-expressive model (2) while preserving the intrinsic dimension of the representation space. Other than using the rank, which is a common measure of the dimension, inspired by (Fazel et al., 2003; Ma et al., 2007; Yu et al., 2020; Liu et al., 2022), we propose to prevent the feature space collapse by incorporating the $\log \det(\cdot)$-based concave smooth surrogate which is defined as follows:

$$R(\boldsymbol{Z}; \alpha) \coloneqq \log \det(\boldsymbol{I} + \alpha \boldsymbol{Z}^\top \boldsymbol{Z}), \tag{3}$$

where $\alpha > 0$ is the hyper-parameter. Unlike the commonly used nuclear norm, which is a convex surrogate of the rank, the $\log \det(\cdot)$-based function is concave and differentiable, offers a tighter approximation and encourages learning subspaces with maximal intrinsic dimensions.[3]

By incorporating the maximization of $R(\boldsymbol{Z}; \alpha)$ as a regularizer into the formulation of SEDSC in (2), we have a Principled fRamewOrk for Deep Subspace Clustering (PRO-DSC):

$$\min_{\boldsymbol{Z},\boldsymbol{C}} \quad -\frac{1}{2}\log\det\left(\boldsymbol{I} + \alpha\boldsymbol{Z}^\top\boldsymbol{Z}\right) + \frac{\gamma}{2}\|\boldsymbol{Z} - \boldsymbol{Z}\boldsymbol{C}\|_F^2 + \beta \cdot r(\boldsymbol{C}) \quad \text{s.t.} \quad \|\boldsymbol{Z}\|_F^2 = N, \tag{4}$$

where $\gamma > 0$ is a hyper-parameter. Now, we will give our theoretical findings for problem (4).

**Theorem 1** (Eigenspace Alignment). *Denote the optimal solution of PRO-DSC in (4) as $(\boldsymbol{Z}_\star, \boldsymbol{C}_\star)$, $\boldsymbol{G}_\star \coloneqq \boldsymbol{Z}_\star^\top \boldsymbol{Z}_\star$ and $\boldsymbol{M}_\star \coloneqq (\boldsymbol{I} - \boldsymbol{C}_\star)(\boldsymbol{I} - \boldsymbol{C}_\star)^\top$. Then $\boldsymbol{G}_\star$ and $\boldsymbol{M}_\star$ share eigenspaces, i.e., $\boldsymbol{G}_\star$ and $\boldsymbol{M}_\star$ can be diagonalized simultaneously by $\boldsymbol{U} \in \mathcal{O}(N)$ where $\mathcal{O}(N)$ is an orthogonal group.*

Note that Theorem 1 provides a perspective from eigenspace alignment for analyzing the property of the optimal solution. Figure 1(a) and (b) show empirical evidences to demonstrate that alignment occurs during the training period, where $\boldsymbol{G}_b = \boldsymbol{Z}_b^\top \boldsymbol{Z}_b$, $\boldsymbol{M}_b = (\boldsymbol{I} - \boldsymbol{C}_b)(\boldsymbol{I} - \boldsymbol{C}_b)^\top$, $\boldsymbol{Z}_b \in \mathbb{R}^{d \times n_b}$, $\boldsymbol{C}_b \in \mathbb{R}^{n_b \times n_b}$ and $n_b$ is batch size, are computed in mini-batch training at different epoch.

---

[1] Without loss of generality, we omit the constraint $\text{diag}(\boldsymbol{C}) = \boldsymbol{0}$ throughout the analysis.

[2] The dimension equals to the multiplicity of the smallest eigenvalues of $(\boldsymbol{I} - \boldsymbol{C})(\boldsymbol{I} - \boldsymbol{C})^\top$.

[3] Please refer to (Ma et al., 2007) for a packing-ball interpretation.

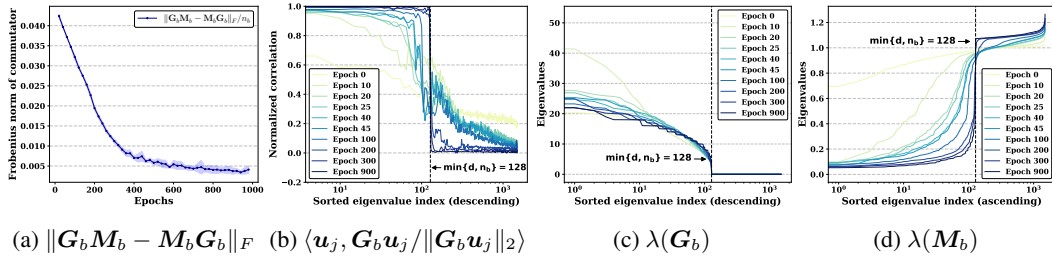

(a) $\|\boldsymbol{G}_b\boldsymbol{M}_b - \boldsymbol{M}_b\boldsymbol{G}_b\|_F$ (b) $\langle\boldsymbol{u}_j, \boldsymbol{G}_b\boldsymbol{u}_j/\|\boldsymbol{G}_b\boldsymbol{u}_j\|_2\rangle$ (c) $\lambda(\boldsymbol{G}_b)$ (d) $\lambda(\boldsymbol{M}_b)$

Figure 1: **Empirical Validation to Eigenspace Alignment and Noncollapse Representation in Mini-batch on CIFAR-100.** (a): Alignment error curve during the training period. (b): Eigenspace correlation curves measured via $\langle\boldsymbol{u}_j, \frac{\boldsymbol{G}_b\boldsymbol{u}_j}{\|\boldsymbol{G}_b\boldsymbol{u}_j\|_2}\rangle$ for $j = 1, \cdots, n_b$. (c) and (d): Eigenvalue curves.

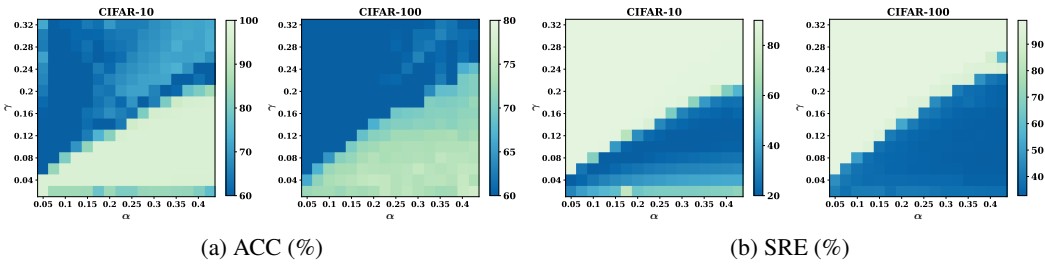

(a) ACC (%)          (b) SRE (%)

Figure 2: **Empirical Validation to Noncollapse Representation on CIFAR-10 and CIFAR-100.** Clustering accuracy (ACC%) and subspace-preserving representation error (SRE%) are displayed under varying $\alpha$ and $\gamma$. When collapse occurs, both ACC and SRE dramatically degenerate. The perceivable phase transition phenomenon is consistent with the condition to avoid collapse.

Next, we will analyze problem (4) from the perspective of alternating optimization. When $\boldsymbol{Z}$ is fixed, the optimization problem with respect to (w.r.t.) $\boldsymbol{C}$ reduces to a standard self-expressive model, which has been extensively studied in (Soltanolkotabi & Candes, 2012; Pimentel-Alarcon & Nowak, 2016; Wang & Xu, 2016; Li et al., 2018; Tsakiris & Vidal, 2018). On the other hand, when $\boldsymbol{C}$ is fixed, the optimization problem w.r.t. $\boldsymbol{Z}$ becomes:

$$\min_{\boldsymbol{Z}} \quad -\frac{1}{2}\log\det\left(\boldsymbol{I} + \alpha\boldsymbol{Z}^\top\boldsymbol{Z}\right) + \frac{\gamma}{2}\|\boldsymbol{Z} - \boldsymbol{Z}\boldsymbol{C}\|_F^2 \quad \text{s.t.} \quad \|\boldsymbol{Z}\|_F^2 = N, \tag{5}$$

which is a *non-convex* optimization problem, whose optimal solution remains under-explored.

In light of the fact that $\boldsymbol{G}$ and $\boldsymbol{M}$ converge to share eigenspaces, we decompose $\boldsymbol{G}$ and $\boldsymbol{M}$ to $\boldsymbol{U}\text{Diag}(\lambda_{\boldsymbol{G}}^{(1)}, \cdots, \lambda_{\boldsymbol{G}}^{(N)})\boldsymbol{U}^\top$ and $\boldsymbol{U}\text{Diag}(\lambda_{\boldsymbol{M}}^{(1)}, \cdots, \lambda_{\boldsymbol{M}}^{(N)})\boldsymbol{U}^\top$, respectively. Recall that $\boldsymbol{G} := \boldsymbol{Z}^\top\boldsymbol{Z}, \boldsymbol{M} := (\boldsymbol{I} - \boldsymbol{C})(\boldsymbol{I} - \boldsymbol{C})^\top$, by using the eigenvalue decomposition, we reformulate problem (5) into a *convex* problem w.r.t. $\{\lambda_{\boldsymbol{G}}^{(i)}\}_{i=1}^{\min\{d,N\}}$ (See Appendix A) and have the following result.

**Theorem 2** (Noncollapse Representation). *Suppose that $\boldsymbol{G}$ and $\boldsymbol{M}$ are aligned in the same eigenspaces and $\gamma < \frac{1}{\lambda_{\max}(\boldsymbol{M})}\frac{\alpha^2}{\alpha+\min\{\frac{d}{N},1\}}$. Then we have: a) $\text{rank}(\boldsymbol{Z}_\star) = \min\{d, N\}$, and b) the singular values $\sigma_{\boldsymbol{Z}_\star}^{(i)} = \sqrt{\frac{1}{\gamma\lambda_{\boldsymbol{M}}^{(i)}+\nu_\star} - \frac{1}{\alpha}}$ for all $i = 1, \ldots, \min\{d, N\}$, where $\boldsymbol{Z}_\star$ and $\nu_\star$ are the optimal primal solution and dual solution, respectively.*

Theorem 2 characterizes the optimal solution for problem (5). Recall that SEDSC in (2) yields a collapsed solution, where $\text{rank}(\boldsymbol{Z}_\star) \ll \min\{d, N\}$; whereas the rank of the minimizers for PRO-DSC in (5) satisfies that $\text{rank}(\boldsymbol{Z}_\star) = \min\{d, N\}$. In Figure 1(c) and (d), we show the curves of the eigenvalues of $\boldsymbol{G}_b$ and $\boldsymbol{M}_b$, which are computed in the mini-batch training at different epoch, demonstrating that the learned representation does no longer collapse. In Figure 2, we show the subspace clustering accuracy (ACC) and subspace-representation error[4] (SRE) as a function of the parameters $\alpha$ and $\gamma$. The phase transition phenomenon around $\gamma < \frac{1}{\lambda_{\max}(\boldsymbol{M})}\frac{\alpha^2}{\alpha+\min\{\frac{d}{N},1\}}$ well illustrates the sufficient condition in Theorem 2 to avoid representation collapse.

---

[4]For each column $\boldsymbol{c}_j$ in $\boldsymbol{C}$, SRE is computed by $\frac{100}{N}\sum_j(1 - \sum_i w_{ij}\cdot|c_{ij}|)/\|\boldsymbol{c}_j\|_1$, where $w_{ij} \in \{0, 1\}$ is the ground-truth affinity.

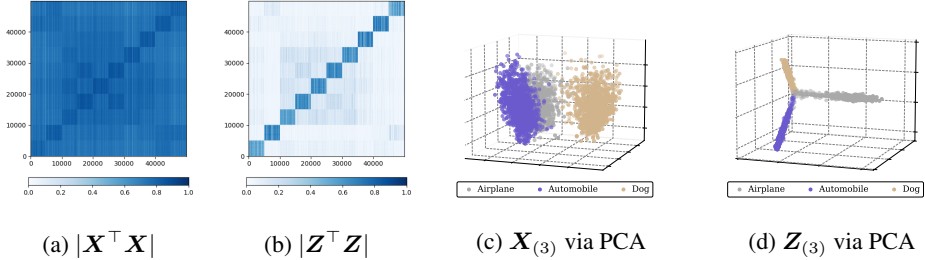

(a) $|\boldsymbol{X}^\top \boldsymbol{X}|$     (b) $|\boldsymbol{Z}^\top \boldsymbol{Z}|$     (c) $\boldsymbol{X}_{(3)}$ via PCA     (d) $\boldsymbol{Z}_{(3)}$ via PCA

Figure 3: **Empirical Validation to Structured Representation on CIFAR-10.** Gram matrices for CLIP features $\boldsymbol{X}$ and learned representations $\boldsymbol{Z}$ are shown in (a) and (b); whereas Data visualization of the samples from three categories $\boldsymbol{X}_{(3)}$ and $\boldsymbol{Z}_{(3)}$ via PCA are shown in (c) and (d), respectively.

Furthermore, from the perspective of joint optimizing $\boldsymbol{Z}$ and $\boldsymbol{C}$, the following theorem demonstrates that PRO-DSC promotes a union-of-orthogonal-subspaces representation $\boldsymbol{Z}$ and block-diagonal self-expressive matrix $\boldsymbol{C}$ under certain condition.

**Theorem 3.** *Suppose that the sufficient conditions to prevent feature collapse are satisfied. Without loss of generality, we further assume that the columns in matrix $\boldsymbol{Z}$ are arranged into $k$ blocks according to a certain $N \times N$ permutation matrix $\boldsymbol{\Gamma}$, i.e., $\boldsymbol{Z} = [\boldsymbol{Z}_1, \boldsymbol{Z}_2, \cdots, \boldsymbol{Z}_k]$. Then the condition for that PRO-DSC promotes the optimal solution $(\boldsymbol{Z}_\star, \boldsymbol{C}_\star)$ to have desired structure (i.e., $\boldsymbol{Z}_\star^\top \boldsymbol{Z}_\star$ and $\boldsymbol{C}_\star$ are both block-diagonal), is that $\langle (\boldsymbol{I} - \boldsymbol{C})(\boldsymbol{I} - \boldsymbol{C})^\top, \boldsymbol{G} - \boldsymbol{G}^* \rangle \to 0$, where $\boldsymbol{G}^* := \mathrm{Diag}\left(\boldsymbol{G}_{11}, \boldsymbol{G}_{22}, \cdots, \boldsymbol{G}_{kk}\right)$ and $\boldsymbol{G}_{jj}$ is the block Gram matrix corresponding to $\boldsymbol{Z}_j$.*

**Remark 1.** Theorem 3 suggests that our PRO-DSC is able to promote learning representations and self-expressive matrix with desired structures, i.e., the representations form a union of orthogonal subspaces and accordingly the self-expressive matrix is block-diagonal, when the condition $\langle (\boldsymbol{I} - \boldsymbol{C})(\boldsymbol{I} - \boldsymbol{C})^\top, \boldsymbol{G} - \boldsymbol{G}^* \rangle \to 0$ is met. We call this condition *a compatibly structured coherence* (CSC), which relates to the properties of the distribution of the representations in $\boldsymbol{Z}$ and the self-coefficients in $\boldsymbol{C}$. While it is not possible for us to give a theoretical justification when the CSC condition will be satisfied in general, we do have the empirical evidence that our implementation for PRO-DSC with careful designs does approximately satisfy such a condition and thus yields representations and self-expressive matrix with desired structure (See Figure 3).[5]

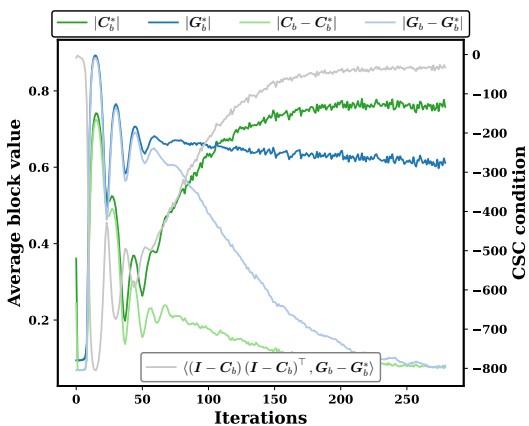

Figure 4: **Empirical validation to Theorem 3 in Mini-batch on CIFAR-10.** The mean curves of the absolute values of the in-block-diagonal entries (thick) and the off-block-diagonal entries (thin) are displayed along with the CSC condition (gray) during training PRO-DSC.

In Figure 4, we show the curves for the compatibly structured coherence (CSC) condition, and for the average values of the entries in $|\boldsymbol{G}_b^*|$, $|\boldsymbol{G}_b - \boldsymbol{G}_b^*|$, $|\boldsymbol{C}_b^*|$, $|\boldsymbol{C}_b - \boldsymbol{C}_b^*|$ computed in mini-batch during training PRO-DSC on CIFAR-10. As illustrated, the CSC condition is progressively satisfied and consequently the average off-block values $|\boldsymbol{G}_b - \boldsymbol{G}_b^*|$ and $|\boldsymbol{C}_b - \boldsymbol{C}_b^*|$ gradually decrease, while the average in-block values $|\boldsymbol{G}_b^*|$ and $|\boldsymbol{C}_b^*|$ gradually increase, which empirically validates that PRO-DSC promotes block-diagonal $\boldsymbol{G}_b$ and $\boldsymbol{C}_b$.

---

[5]Please refer to Appendix B.2 for more details about Figures 1-4.

## 2.3 Scalable Implementation

Existing SEDSC models typically use autoencoders to learn the representations and learn the self-expressive matrix $C$ through an $N \times N$ fully-connected layer (Peng et al., 2016; 2018; Ji et al., 2017; Zhou et al., 2018; Zhang et al., 2019a). While such implementation is straightforward, there are two major drawbacks: a) since that the number of self-expressive coefficients is quadratic to the number of data points, solving these coefficients suffers from expensive computation burden; b) the learning process is transductive, i.e., the network parameters cannot be generalized to unseen data.

To address these issues, similar to (Zhang et al., 2021), we reparameterize the self-expressive coefficients $c_{ij}$ by a neural network. Specifically, the input data $x_i$ is fed into a neural network $h(\cdot; \Psi) : \mathbb{R}^D \to \mathbb{R}^d$ to yield normalized representations, i.e.,

$$y_i := h(x_i; \Psi)/\|h(x_i; \Psi)\|_2, \tag{6}$$

where $\Psi$ denotes all the parameters in $h(\cdot)$. Then, the parameterized self-expressive matrix $C_{\Psi}$ is generated by:

$$C_{\Psi} := \mathcal{P}(Y^\top Y), \tag{7}$$

where $Y := [y_1, \ldots, y_N] \in \mathbb{R}^{d \times N}$ and $\mathcal{P}(\cdot)$ is the sinkhorn projection (Cuturi, 2013), which has been widely applied in deep clustering (Caron et al., 2020; Ding et al., 2023).[6] To enable efficient representation learning, we introduce another learnable mapping $f(\cdot; \Theta) : \mathbb{R}^D \to \mathbb{R}^d$, for which

$$z_j := f(x_j; \Theta)/\|f(x_j; \Theta)\|_2 \tag{8}$$

is the learned representation for the input $x_j$, where $\Theta$ denotes the parameters in $f(\cdot)$ to learn the structured representation $Z_{\Theta} := [z_1, \ldots, z_N] \in \mathbb{R}^{d \times N}$.

Therefore, our principled framework for deep subspace clustering (PRO-DSC) in (4) can be reparameterized and reformulated as follows:

$$\min_{\Theta, \Psi} \quad \mathcal{L}(\Theta, \Psi) := -\frac{1}{2} \log \det \left( I + \alpha Z_{\Theta}^\top Z_{\Theta} \right) + \frac{\gamma}{2} \|Z_{\Theta} - Z_{\Theta} C_{\Psi}\|_F^2 + \beta \cdot r(C_{\Psi}). \tag{9}$$

To strengthen the block-diagonal structure of self-expressive matrix, we choose the block-diagonal regularizer (Lu et al., 2018) for $r(C_{\Psi})$. To be specific, given the data affinity $A_{\Psi}$, which is induced by default as $A_{\Psi} := \frac{1}{2} \left( |C_{\Psi}| + |C_{\Psi}^\top| \right)$, the block diagonal regularizer is defined as:

$$r(C_{\Psi}) := \|A_{\Psi}\|_{\boxed{\kappa}}, \tag{10}$$

where $\|A_{\Psi}\|_{\boxed{\kappa}}$ is the sum of the $k$ smallest eigenvalues of the Laplacian matrix of the affinity $A_{\Psi}$.[7]

Consequently, the parameters in $\Theta$ and $\Psi$ of reparameterized PRO-DSC can be trained by Stochastic Gradient Descent (SGD) with the loss function $\mathcal{L}(\Theta, \Psi)$ defined in (9). For clarity, we summarize the procedure for training and testing of our PRO-DSC in Algorithm 1.

**Remark 2.** We note that all the commonly used regularizers with extended block-diagonal property for self-expressive model as discussed in (Lu et al., 2018) can be used to improve the block-diagonal structure of self-expressive matrix. More interestingly, the specific type of the regularizers is not essential owning to the learned structured representation (Please refer to Table 3 for details), and using a specific regularizer or not is also not essential since that the SGD-based optimization also induces some implicit regularization, e.g., low-rank (Gunasekar et al., 2017; Arora et al., 2019).

## 3 Experiments

To validate our theoretical findings and to demonstrate the performance of our proposed framework, we conduct extensive experiments on synthetic data (Sec. 3.1) and real-world data (Sec. 3.2). Implementation details and more results are provided in Appendices B.1 and B.3, respectively.

---

[6] In practice, we set $\mathrm{diag}(C_{\Psi}) = 0$ to prevent trivial solution $C_{\Psi} = I$.

[7] Recall that the number of zero eigenvalues of the Laplacian matrix equals to the number of connected components in the graph (von Luxburg, 2007).

---

**Algorithm 1** Scalable & Efficient Implementation of PRO-DSC via Differential Programming

---

**Input:** Dataset $\mathcal{X} = \mathcal{X}_{\text{train}} \cup \mathcal{X}_{\text{test}}$, batch size $n_b$, hyper-parameters $\alpha, \beta, \gamma$, number of iterations $T$, learning rate $\eta$

**Initialization:** Random initialize the parameters $\boldsymbol{\Psi}, \boldsymbol{\Theta}$ in the networks $\boldsymbol{h}(\cdot; \boldsymbol{\Psi})$ and $\boldsymbol{f}(\cdot; \boldsymbol{\Theta})$

**Training:**
  1: **for** $t = 1, \ldots, T$ **do**
  2:      Sample a batch $\boldsymbol{X}_b \in \mathbb{R}^{D \times n_b}$ from $\mathcal{X}_{\text{train}}$
            *# Forward propagation*
  3:      Compute self-expressive matrix $\boldsymbol{C}_b \in \mathbb{R}^{n_b \times n_b}$ by Eqs. (6–7)
  4:      Compute representations $\boldsymbol{Z}_b \in \mathbb{R}^{d \times n_b}$ by Eq. (8)
            *# Backward propagation*
  5:      Compute gradients: $\nabla_{\boldsymbol{\Psi}} := \frac{\partial \mathcal{L}}{\partial \boldsymbol{\Psi}}, \nabla_{\boldsymbol{\Theta}} := \frac{\partial \mathcal{L}}{\partial \boldsymbol{\Theta}}$
  6:      Update $\boldsymbol{\Psi}$ and $\boldsymbol{\Theta}$ via: $\boldsymbol{\Psi} \leftarrow \boldsymbol{\Psi} - \eta \cdot \nabla_{\boldsymbol{\Psi}}, \boldsymbol{\Theta} \leftarrow \boldsymbol{\Theta} - \eta \cdot \nabla_{\boldsymbol{\Theta}}$
  7: **end for**
**Testing:**
  8: Compute self-expressive matrix $\boldsymbol{C}_{\text{test}}$ by Eqs. (6–7) for $\mathcal{X}_{\text{test}}$
  9: Apply spectral clustering on the affinity $\boldsymbol{A}_{\text{test}}$

---

### 3.1 EXPERIMENTS ON SYNTHETIC DATA

To validate whether PRO-DSC resolves the collapse issue in SEDSC and learns representations with a UoS structure, we first follow the procedure in (Ding et al., 2023) to generate two sets of synthetic data, as shown in the first column of Figure 5, and then visualize in Figure 5(b)-(e) the learned representations which are obtained from different methods on these synthetic data.

We observe that the SEDSC model overly compress all the representations to a closed curve on the hypersphere; whereas with increased weights (i.e., $\gamma \uparrow$) of the self-expressive term, the representations collapse to a few points. Our PRO-DSC yields linearized representations lying on orthogonal subspaces in both cases, confirming the effectiveness of our approach. Nevertheless, MLC (Ding et al., 2023) yields representations approximately on orthogonal subspaces.

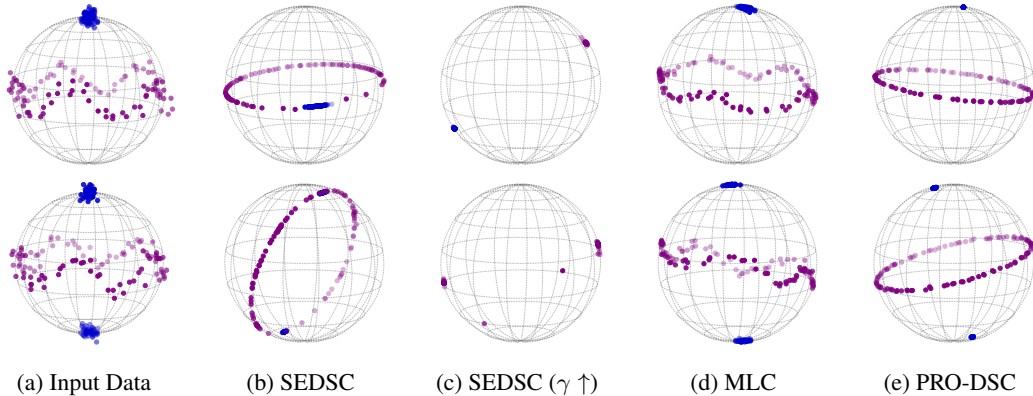

|    (a) Input Data    |    (b) SEDSC    |    (c) SEDSC ($\gamma \uparrow$)    |    (d) MLC    |    (e) PRO-DSC    |

Figure 5: Visualization Experiments on Synthetic Data.

### 3.2 EXPERIMENTS ON REAL-WORLD DATA

To evaluate the performance of our proposed approach, we conduct experiments on six real-world image datasets, including CIFAR-10, CIFAR-20, CIFAR-100, ImageNet-Dogs-15, Tiny-ImageNet-200, and ImageNet-1k, with the pretrained CLIP features[8] (Radford et al., 2021), and compare to several baseline methods, including classical clustering algorithms, e.g., $k$-means (MacQueen, 1967) and spectral clustering (Shi & Malik, 2000), subspace clustering algorithm, e.g., EnSC (You et al., 2016a) and SENet (Zhang et al., 2021), deep clustering algorithms, e.g., SCAN (Van Gansbeke et al., 2020), TEMI (Adaloglou et al., 2023) and CPP (Chu et al., 2024), and deep subspace clustering algorithms, e.g., DSCNet (Ji et al., 2017) and EDESC (Cai et al., 2022). We measure clustering

---

[8]Please refer to Appendix B.3 for the results on other pre-trained models.

Table 1: **Clustering performance Comparison on the CLIP features.** The best results are in bold and the second best results are underlined. "OOM" means out of GPU memory.

| Method | CIFAR-10 | | CIFAR-20 | | CIFAR-100 | | TinyImgNet-200 | | ImgNetDogs-15 | | ImageNet-1k | |
|---|---|---|---|---|---|---|---|---|---|---|---|---|
| | ACC | NMI | ACC | NMI | ACC | NMI | ACC | NMI | ACC | NMI | ACC | NMI |
| $k$-means | 83.5 | 84.1 | 46.9 | 49.4 | 52.8 | 66.8 | 54.1 | 73.4 | 52.7 | 53.6 | 53.9 | 79.8 |
| SC | 79.8 | 84.8 | 53.3 | 61.6 | 66.4 | 77.0 | 62.8 | 77.0 | 48.3 | 45.7 | 56.0 | 81.2 |
| SSCOMP | 85.5 | 83.0 | 61.4 | 63.4 | 55.6 | 69.7 | 56.7 | 72.7 | 25.6 | 15.9 | 44.1 | 74.4 |
| EnSC | 95.4 | 90.3 | 61.0 | 68.7 | 67.0 | 77.1 | 64.5 | 77.7 | 57.9 | 56.0 | 59.7 | **83.7** |
| SENet | 91.2 | 82.5 | 65.3 | 68.6 | 67.0 | 74.7 | 63.9 | 76.6 | 58.7 | 55.3 | 53.2 | 78.1 |
| SCAN | 95.1 | 90.3 | 60.8 | 61.8 | 64.1 | 70.8 | 56.5 | 72.7 | 70.5 | 68.2 | 54.4 | 76.8 |
| TEMI | 96.9 | 92.6 | 61.8 | 64.5 | 73.7 | 79.9 | - | - | - | - | 64.0 | - |
| CPP | 96.8 | 92.3 | 67.7 | 70.5 | 75.4 | 82.0 | 63.4 | 75.5 | 83.0 | **81.5** | 62.0 | 82.1 |
| EDESC | 84.2 | 79.3 | 48.7 | 49.1 | 53.1 | 68.6 | 51.3 | 68.8 | 53.3 | 47.9 | 46.5 | 75.5 |
| DSCNet | 78.5 | 73.6 | 38.6 | 45.7 | 39.2 | 53.4 | 62.3 | 68.3 | 40.5 | 30.1 | OOM | OOM |
| Our PRO-DSC | **97.2**±0.2 | **92.8**±0.4 | **71.6**±1.2 | **73.2**±0.5 | **77.3**±1.0 | **82.4**±0.5 | **69.8**±1.1 | **80.5**±0.7 | **84.0**±0.6 | 81.2±0.8 | **65.0**±1.2 | 83.4±0.6 |

performance using clustering accuracy (ACC) and normalized mutual information (NMI), and report the experimental results in Table 1, where the results of our PRO-DSC are averaged over 10 trials (with ±std). Since that for most baselines, except for TEMI, the clustering performance with the CLIP feature has not been reported, we conduct experiments using the implementations provided by the authors. For TEMI, we directly cited the results from (Adaloglou et al., 2023).

**Performance comparison.** As shown in Table 1, our PRO-DSC significantly outperforms subspace clustering algorithms, e.g., SSCOMP, EnSC and SENet, and deep subspace clustering algorithms, e.g., DSCNet and EDESC. Moreover, our PRO-DSC obtains better performance than the state-of-the-art deep clustering and deep manifold clustering methods, e.g., SCAN, TEMI and CPP.

**Validation to the theoretical results.** To validate whether the alignment emerges and when representations collapse occurs during the training period, we compute $G_b = Z_b^\top Z_b$ and $M_b = (I - C_b)(I - C_b)^\top$ in mini-batch at different epoch during the training period, and then measure the alignment error via $\|G_b M_b - M_b G_b\|_F$ and the eigenspace correlation via $\langle u_j, \frac{G_b u_j}{\|G_b u_j\|_2}\rangle$ where $u_j$ is the $j$-th ending eigenvector[9] of $M_b$ for $j = 1, \cdots, n_b$, and plot the eigenvalues of $G_b$ and $M_b$, where $n_b$ is the sample size per mini-batch. Moreover, we also record empirical performance ACC and SRE on CIFAR-10 and CIFAR-100 under varying hyper-parameters $\alpha$ and $\gamma$ to validate the condition in Theorem 2 to avoid collapse. Experimental results are displayed in Figures 1 and 2. We observe that $G_b$ and $M_b$ are increasingly aligned and the representations will no longer collapse provided that the parameters are properly set. More details are provided in Section B.2.

**Evaluation on learned representations.** To quantitatively evaluate the effectiveness of the learned representations, we run $k$-means (MacQueen, 1967), spectral clustering (Shi & Malik, 2000), and EnSC (You et al., 2016a) on four datasets with three different features: a) the CLIP features, b) the representations learned via CPP, and c) the representations learned by our PRO-DSC. Experimental results are shown in Figure 6 (and more results are given in Table B.4 of Appendix B.3). We observe that the representations learned by our PRO-DSC outperform the CLIP features and the CPP representations in most cases across different clustering algorithms and datasets. Notably, the clustering accuracy with the representations learned by our PRO-DSC exceeds 90% on CIFAR-10 and 75% on CIFAR-100, whichever clustering algorithm is used. Besides, the clustering performance is further improved by using the learnable mapping $h(\cdot; \Psi)$, indicating a good generalization ability.

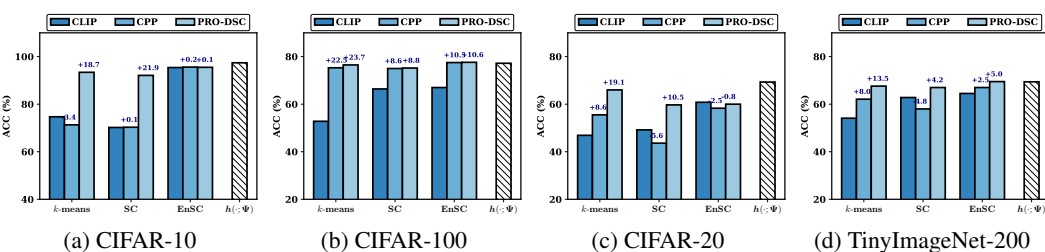

Figure 6: Clustering accuracy with CLIP features and learned representations.

[9]The eigenvectors are sorted according to eigenvalues of $M_b$ in ascending order.

**Sensitivity to hyper-parameters.** In Figure 2, we verify that our PRO-DSC yields satisfactory results when the conditions in Theorem 2 to avoid collapse are met. Moreover, we evaluate the performance sensitivity to hyper-parameters $\gamma$ and $\beta$ by experiments on the CLIP features of CIFAR-10, CIFAR-100 and TinyImageNet-200 with varying $\gamma$ and $\beta$. In Figure 7, we observe that the clustering performance maintains satisfactory under a broad range of $\gamma$ and $\beta$.

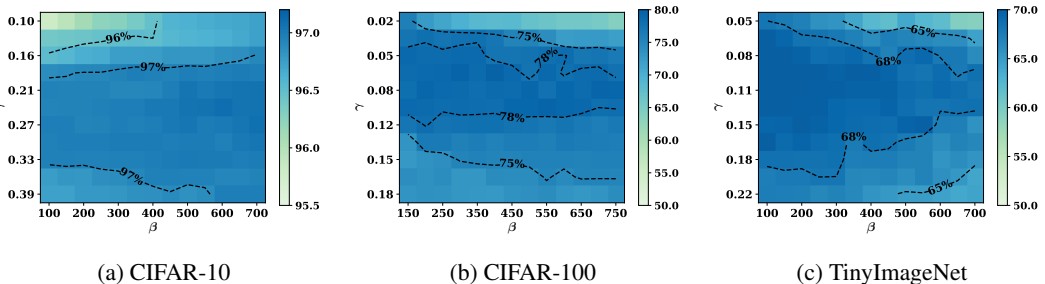

| (a) CIFAR-10 | (b) CIFAR-100 | (c) TinyImageNet |
|---|---|---|

Figure 7: Evaluation on sensitivity to hyper-parameters $\gamma$ and $\beta$ on three datasets.

**Time and memory cost.** The most time-consuming operations in our PRO-DSC are computing the term involving $\log \det(\cdot)$ and the term $\|\boldsymbol{A}\|_{\boxed{\kappa}}$ involving eigenvalue decomposition, respectively. The time complexity for $\log \det(\cdot)$ is $\mathcal{O}(\min\{n_b^3, d^3\})$ due to the commutative property of $\log \det(\cdot)$ function (Yu et al., 2020), and the time complexity for $\|\boldsymbol{A}\|_{\boxed{\kappa}}$ is $\mathcal{O}(kn_b^2)$.[10] Therefore the overall time complexity of our PRO-DSC is $\mathcal{O}(kn_b^2 + \min\{n_b^3, d^3\})$. Note that TEMI (Adaloglou et al., 2023) employs $H = 50$ cluster heads during training, adding further time and memory costs and CPP (Chu et al., 2024) involves computing $\log \det(\cdot)$ for $n_b + 1$ times, leading to complexity $\mathcal{O}((n_b + 1) \min\{n_b^3, d^3\})$. The computation time and memory costs are shown in Table 2 for which all the experiments are conducted on a single NVIDIA RTX 3090 GPU and Intel Xeon Platinum 8255C CPU. We read that our PRO-DSC significantly reduces the time consumption, particularly for datasets with a large number of clusters.

Table 2: Comparison on time (s) and memory cost (MiB). "OOM" means out of GPU memory.

| Methods | Complexity | CIFAR-10 Time | CIFAR-10 Memory | CIFAR-100 Time | CIFAR-100 Memory | ImageNet-1k Time | ImageNet-1k Memory |
|---|---|---|---|---|---|---|---|
| SEDSC | $\mathcal{O}(N^2 d)$ | - | OOM | - | OOM | - | OOM |
| TEMI | $\mathcal{O}(H n_b d^2)$ | 6.9 | **1,766** | 5.1 | 2,394 | 262.1 | 2,858 |
| CPP | $\mathcal{O}((n_b + 1) \min\{n_b^3, d^3\})$ | **3.5** | 3,802 | 7.1 | 10,374 | 1441.2 | 22,433 |
| PRO-DSC | $\mathcal{O}(k n_b^2 + \min\{n_b^3, d^3\})$ | 4.5 | 2,158 | **4.0** | **2,328** | **90.0** | **2,335** |

Table 3: Ablation studies on different loss functions and regularizers.

| | | $\mathcal{L}_1$ | $\mathcal{L}_2$ | $\|\boldsymbol{A}\|_{\boxed{\kappa}}$ | $\|\boldsymbol{C}\|_1$ | $\|\boldsymbol{C}\|_F^2$ | $\|\boldsymbol{C}\|_*$ | CIFAR-10 ACC | CIFAR-10 NMI | CIFAR-100 ACC | CIFAR-100 NMI | ImgNetDogs-15 ACC | ImgNetDogs-15 NMI |
|---|---|---|---|---|---|---|---|---|---|---|---|---|---|
| Ablation | | | ✓ | ✓ | | | | 56.9 | 47.7 | 54.6 | 60.9 | 46.7 | 37.1 |
| | | ✓ | | ✓ | | | | 69.6 | 56.4 | 64.7 | 71.7 | 10.5 | 1.7 |
| | | ✓ | ✓ | | | | | 97.0 | **93.0** | 74.6 | 80.9 | 80.9 | 78.8 |
| Regularizer | | ✓ | ✓ | | | | ✓ | 97.0 | 92.6 | 75.2 | 81.1 | 81.3 | 79.1 |
| | | ✓ | ✓ | | | ✓ | | 97.0 | 92.6 | 75.2 | 80.9 | 80.9 | 78.8 |
| | | ✓ | ✓ | | ✓ | | | 96.7 | 91.9 | 76.4 | 81.8 | 81.0 | 78.8 |
| | | ✓ | ✓ | ✓ | | | | **97.2** | 92.8 | **77.3** | **82.4** | **84.0** | **81.2** |

**Ablation study.** To verify the effectiveness of each components in the loss function of our PRO-DSC, we conduct a set of ablation studies with the CLIP features on CIFAR-10, CIFAR-100, and ImageNetDogs-15, and report the results in Table 3, where $\mathcal{L}_1 := -\frac{1}{2} \log \det \left( \boldsymbol{I} + \alpha \boldsymbol{Z}_{\boldsymbol{\Theta}}^\top \boldsymbol{Z}_{\boldsymbol{\Theta}} \right)$ and $\mathcal{L}_2 := \frac{1}{2} \|\boldsymbol{Z}_{\boldsymbol{\Theta}} - \boldsymbol{Z}_{\boldsymbol{\Theta}} \boldsymbol{C}_{\boldsymbol{\Psi}}\|_F^2$. The absence of the term $\mathcal{L}_1$ leads to catastrophic feature collapse (as demonstrated in Sec. 2.1); whereas without the self-expressive $\mathcal{L}_2$, the model lacks a loss function

---

[10]For an $N \times N$ matrix, the complexity of computing its $k$ eigenvalues by Lanczos algorithm is $\mathcal{O}(kN^2)$ and the complexity of computing its $\det(\cdot)$ is $\mathcal{O}(N^3)$.

for learning the self-expressive coefficients. In both cases, clustering performance drops significantly. More interestingly, when we replace the block diagonal regularizer $\|\boldsymbol{A}\|_{\overline{K}}$ with $\|\boldsymbol{C}\|_1$, $\|\boldsymbol{C}\|_*$, and $\|\boldsymbol{C}\|_F^2$ or even drop the explicit regularizer $r(\cdot)$, the clustering performance still maintains satisfactory. This confirms that the choice of the regularizer is not essential owning to the structured representations learned by our PRO-DSC.

## 4 RELATED WORK

**Deep subspace clustering.** To tackle with complex real world data, a number of Self-Expressive Deep Subspace Clustering (SEDSC) methods have been developed in the past few years, e.g., (Peng et al., 2016; 2018; Ji et al., 2017; Zhou et al., 2018; Zhang et al., 2019a;b; Dang et al., 2020; Peng et al., 2020; Lv et al., 2021; Cai et al., 2022; Wang et al., 2023b). The key step in SEDSC is to adopt a deep learning module to embed the input data into feature space. For example, deep autoencoder network is adopted in (Peng et al., 2016; 2018), deep convolutional autoencoder network is used in (Ji et al., 2017; Zhou et al., 2018; Zhang et al., 2019a). Unfortunately, as pointed out in (Haeffele et al., 2021), the existing SEDSC methods suffer from a catastrophic feature collapse and there is no evidence that the learned representations align with a UoS structure. To date, however, a principled deep subspace clustering framework has not been proposed.

**Deep clustering.** Recently, most of state-of-the-art deep clustering methods adopt a two-step procedure: at the first step, self-supervised learning based pre-training, e.g., SimCLR (Chen et al., 2020a), MoCo (He et al., 2020), BYOL (Grill et al., 2020) and SwAV (Caron et al., 2020), is adopted to learn the representations; and then deep clustering methods are incorporated to refine the representations, via, e.g., pseudo-labeling (Caron et al., 2018; Van Gansbeke et al., 2020; Park et al., 2021; Niu et al., 2022), cluster-level contrastive learning (Li et al., 2021), local and global neighbor matching (Dang et al., 2021), graph contrastive learning (Zhong et al., 2021), self-distillation (Adaloglou et al., 2023). Though the clustering performance has been improved remarkably, the underlying geometry structure of the learned representations is unclear and ignored.

**Representation learning with a UoS structure.** The methods for representation learning that favor a UoS structure are pioneered in supervised setting, e.g., (Lezama et al., 2018; Yu et al., 2020). In (Lezama et al., 2018), a nuclear norm based geometric loss is proposed to learn representations that lie on a union of orthogonal subspaces; in (Yu et al., 2020), a principled framework called Maximal Coding Rate Reduction (MCR$^2$) is proposed to learn representations that favor the structure of a union of orthogonal subspaces (Wang et al., 2024). More recently, the MCR$^2$ framework is modified to develop deep manifold clustering methods, e.g., NMCE (Li et al., 2022), MLC (Ding et al., 2023) and CPP (Chu et al., 2024). In (Li et al., 2022), the MCR$^2$ framework combines with constrastive learning to perform manifold clustering and representation learning; in (Ding et al., 2023), the MCR$^2$ framework combines with doubly stochastic affinity learning to perform manifold linearizing and clustering; and in (Chu et al., 2024), the features from large pre-trained model (e.g., CLIP) are adopted to evaluate the performance of (Ding et al., 2023). While the MCR$^2$ framework has been modified in these methods for manifold clustering, none of them provides theoretical justification to yield structured representations. Though our PRO-DSC shares the same regularizer defined in Eq. (3) with MLC (Ding et al., 2023), we are for the first time to adopt it into the SEDSC framework to attack the catastrophic feature collapse issue with theoretical analysis.

## 5 CONCLUSION

We presented a Principled fRamewOrk for Deep Subspace Clustering (PRO-DSC), which jointly learn structured representations and self-expressive coefficients. Specifically, our PRO-DSC incorporates an effective regularization into self-expressive model to prevent the catastrophic representation collapse with theoretical justification. Moreover, we demonstrated that our PRO-DSC is able to learn structured representations that form a desirable UoS structure, and also developed an efficient implementation based on reparameterization and differential programming. We conducted extensive experiments on synthetic data and six benchmark datasets to verify the effectiveness of our proposed approach and validate our theoretical findings.

## ACKNOWLEDGMENTS

The authors would like to thank the constructive comments from anonymous reviewers. This work is supported by the National Natural Science Foundation of China under Grant 61876022.

## ETHICS STATEMENT

In this work, we aim to extend traditional subspace clustering algorithms by leveraging deep learning techniques to enhance their representation learning capabilities. Our research does not involve any human subjects, and we have carefully ensured that it poses no potential risks or harms. Additionally, there are no conflicts of interest, sponsorship concerns, or issues related to discrimination, bias, or fairness associated with this study. We have taken steps to address privacy and security concerns, and all data used comply with legal and ethical standards. Our work fully adheres to research integrity principles, and no ethical concerns have arisen during the course of this study.

## REPRODUCIBILITY STATEMENT

To ensure the reproducibility of our work, we have released the source code. Theoretical proofs of the claims made in this paper are provided in Appendix A, and the empirical validation of these theoretical results is shown in Figures 2-4, with further detailed explanations in Appendix B.2. All datasets used in our experiments are publicly available, and we have provided a comprehensive description of the data processing steps in Appendix B.1. Additionally, detailed experimental settings and configurations are outlined in Appendix B.1 to facilitate the reproduction of our results.

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

# SUPPLEMENTARY MATERIAL FOR "EXPLORING A PRINCIPLED FRAMEWORK FOR DEEP SUBSPACE CLUSTERING"

The supplementary materials are divided into three parts. In Section A, we present the proofs of our theoretical results. In Section B, we present the supplementary materials for experiments, including experimental details (Sec. B.1), empirical validation on our theoretical results (Sec. B.2), and more experimental results (Sec. B.3). In Section C, we discuss about the limitations and failure cases of PRO-DSC.

## A PROOFS OF MAIN RESULTS

As a preliminary, we start by introducing a lemma from (Haeffele et al., 2021) and provide its proof for the convenience of the readers.

**Lemma 1** (Haeffele et al., 2021). *The rows of the optimal solution $\boldsymbol{Z}$ for problem (2) are the eigenvectors that associate with the smallest eigenvalues of $(\boldsymbol{I} - \boldsymbol{C})(\boldsymbol{I} - \boldsymbol{C})^\top$.*

*Proof.* We note that:

$$\|\boldsymbol{Z} - \boldsymbol{Z}\boldsymbol{C}\|_F^2 = \mathrm{Tr}\left(\boldsymbol{Z}\left(\boldsymbol{I} - \boldsymbol{C}\right)\left(\boldsymbol{I} - \boldsymbol{C}\right)^\top \boldsymbol{Z}^\top\right) = \sum_{i=1}^d \boldsymbol{z}^{(i)}(\boldsymbol{I} - \boldsymbol{C})(\boldsymbol{I} - \boldsymbol{C})^\top \boldsymbol{z}^{(i)\top},$$

where $\boldsymbol{z}^{(i)}$ is the $i^{\text{th}}$ row of $\boldsymbol{Z}$, thus problem (2) is reformulated as:

$$\min_{\{\boldsymbol{z}^{(i)}\}_{i=1}^d, \boldsymbol{C}} \quad \frac{1}{2}\sum_{i=1}^d \boldsymbol{z}^{(i)}(\boldsymbol{I} - \boldsymbol{C})(\boldsymbol{I} - \boldsymbol{C})^\top \boldsymbol{z}^{(i)\top} + \beta \cdot r(\boldsymbol{C}) \tag{11}$$
$$\text{s.t.} \quad \|\boldsymbol{Z}\|_F^2 = N.$$

Without loss of generality, the magnitude of each row of $\boldsymbol{Z}$ is assumed to be fixed, i.e., $\|\boldsymbol{z}^{(i)}\|_2^2 = \tau_i$, $i = 1, \ldots, d$, where $\sum_{i=1}^d \tau_i = N$. Then, the optimization problem becomes:

$$\min_{\{\boldsymbol{z}^{(i)}\}_{i=1}^d, \boldsymbol{C}} \quad \frac{1}{2}\sum_{i=1}^d \boldsymbol{z}^{(i)}(\boldsymbol{I} - \boldsymbol{C})(\boldsymbol{I} - \boldsymbol{C})^\top \boldsymbol{z}^{(i)\top} + \beta \cdot r(\boldsymbol{C}) \tag{12}$$
$$\text{s.t.} \quad \|\boldsymbol{z}^{(i)}\|_2^2 = \tau_i, \quad i = 1, \ldots, d.$$

The Lagrangian of problem (12) is:

$$\mathcal{L}(\{\boldsymbol{z}^{(i)}\}_{i=1}^d, \boldsymbol{C}, \{\nu_i\}_{i=1}^d) := \frac{1}{2}\sum_{i=1}^d \boldsymbol{z}^{(i)}(\boldsymbol{I} - \boldsymbol{C})(\boldsymbol{I} - \boldsymbol{C})^\top \boldsymbol{z}^{(i)\top} + \beta \cdot r(\boldsymbol{C}) + \frac{1}{2}\sum_{i=1}^d \nu_i(\|\boldsymbol{z}^{(i)}\|_2^2 - \tau_i), \tag{13}$$

where $\{\nu_i\}_{i=1}^d$ are the Lagrangian multipliers. The necessary conditions for optimal solution are:

$$\begin{cases} \nabla_{\boldsymbol{z}^{(i)}}\mathcal{L} = \boldsymbol{z}^{(i)}(\boldsymbol{I} - \boldsymbol{C})(\boldsymbol{I} - \boldsymbol{C})^\top + \nu_i \boldsymbol{z}^{(i)} = \boldsymbol{0}, \\ \|\boldsymbol{z}^{(i)}\|_2^2 = \tau_i, \quad i = 1, \ldots, d, \end{cases} \tag{14}$$

which implies that the optimal solutions $\boldsymbol{z}^{(i)}$ are eigenvectors of $(\boldsymbol{I} - \boldsymbol{C})(\boldsymbol{I} - \boldsymbol{C})^\top$.

By further considering the objective functions, the optimal solution $\boldsymbol{z}^{(i)}$ should be the eigenvectors w.r.t. the *smallest* eigenvalues of $(\boldsymbol{I} - \boldsymbol{C})(\boldsymbol{I} - \boldsymbol{C})^\top$ for all $i \in \{1, \ldots, d\}$. The corresponding optimal value is $\frac{1}{2}\lambda_{\min}((\boldsymbol{I} - \boldsymbol{C})(\boldsymbol{I} - \boldsymbol{C})^\top)\sum_{i=1}^d \tau_i + \beta \cdot r(\boldsymbol{C}) = \frac{N}{2}\lambda_{\min}((\boldsymbol{I} - \boldsymbol{C})(\boldsymbol{I} - \boldsymbol{C})^\top) + \beta \cdot r(\boldsymbol{C})$, which is irrelevant to $\{\tau_i\}_{i=1}^d$.

Therefore, we conclude that the rows of optimal solution $\boldsymbol{Z}$ to problem (2) are eigenvectors that associate with the smallest eigenvalues of $(\boldsymbol{I} - \boldsymbol{C})(\boldsymbol{I} - \boldsymbol{C})^\top$.

$\square$

**Lemma A1.** *Suppose that matrices $A, B \in \mathbb{R}^{n \times n}$ are symmetric, then $AB = BA$ if and only if $A$ and $B$ can be diagonalized simultaneously by $U \in \mathcal{O}(n)$, where $\mathcal{O}(n)$ is an orthogonal group.*

Now we present our theorem about the optimal solution of problem PRO-DSC in (4) with its proof.

**Theorem 1.** *Denote the optimal solution of PRO-DSC in (4) as $(Z_\star, C_\star)$, then $G_\star$ and $M_\star$ share eigenspaces, where $G_\star \coloneqq Z_\star^\top Z_\star, M_\star \coloneqq (I - C_\star)(I - C_\star)^\top$, i.e., $G_\star$ and $M_\star$ can be diagonalized simultaneously by $U \in \mathcal{O}(N)$ where $\mathcal{O}(N)$ is an orthogonal matrix group.*

*Proof.* We first consider the subproblem of PRO-DSC problem in (4) with respect to $Z$ and prove that for all $C \in \mathbb{R}^{N \times N}$, the corresponding optimal $Z_{\star,C}$ satisfies $G_{\star,C}M = MG_{\star,C}$, where $G_{\star,C} = Z_{\star,C}^\top Z_{\star,C}, M \coloneqq (I - C)(I - C)^\top$, implying that $G_{\star,C}$ and $M$ share eigenspace. Then, we will demonstrate that $G_\star$ and $M_\star$ share eigenspace.

The subproblem with respect to $Z_C$ is reformulated into the following semi-definite program:

$$
\begin{aligned}
\min_{G_C} \quad & -\frac{1}{2}\log\det\left(I + \alpha G_C\right) + \frac{\gamma}{2}\operatorname{tr}(G_C M) \\
\text{s.t.} \quad & G_C \succeq 0, \ \operatorname{tr}(G_C) = N,
\end{aligned}
\tag{15}
$$

which has the Lagrangian as:

$$
\mathcal{L}(G_C, \Delta, \nu) \coloneqq -\frac{1}{2}\log\det\left(I + \alpha G_C\right) + \frac{\gamma}{2}\operatorname{tr}(G_C M) - \operatorname{tr}(\Delta G_C) + \frac{\nu}{2}(\operatorname{tr}(G_C) - N), \tag{16}
$$

where scalar $\nu$ and $N \times N$ symmetric matrix $\Delta$ are Lagrange multipliers.

The KKT conditions is:

$$
\begin{cases}
-\frac{\alpha}{2}(I + \alpha G_{\star,C})^{-1} + \frac{\gamma}{2}M - \Delta_\star + \frac{\nu_\star}{2}I = 0, & (17) \\
G_{\star,C} \succeq 0, & (18) \\
\operatorname{tr}(G_{\star,C}) = N, & (19) \\
\Delta_\star \succeq 0, & (20) \\
\Delta_\star G_{\star,C} = 0, & (21)
\end{cases}
$$

which are the sufficient and necessary condition for the global optimality of the solution $G_{\star,C}$.

From Eqs. (18),(20) and (21), we have that $\Delta_\star G_{\star,C} - G_{\star,C}\Delta_\star = \Delta_\star G_{\star,C} - (\Delta_\star G_{\star,C})^\top = 0$, implying that $\Delta_\star$ and $G_{\star,C}$ share eigenspace. By eigenvalue decomposition $\Delta_\star = Q\Lambda_{\Delta_\star}Q^\top, G_{\star,C} = Q\Lambda_{G_{\star,C}}Q^\top$, where $\Lambda_{\Delta_\star}, \Lambda_{G_{\star,C}}$ are diagonal matrices, we have:

$$
2 \cdot Q\Lambda_{\Delta_\star}Q^\top = -\alpha Q(I + \alpha\Lambda_{G_{\star,C}})^{-1}Q^\top + \gamma M + \nu_\star I \tag{22}
$$

$$
\Rightarrow \quad \gamma M + \nu_\star I = Q\left(2\Lambda_{\Delta_\star} + \alpha\left(I + \alpha\Lambda_{G_{\star,C}}\right)^{-1}\right)Q^\top, \tag{23}
$$

where the first equality is from Eq. (17). Since that $2\Lambda_{\Delta_\star} + \alpha(I + \alpha\Lambda_{G_{\star,C}})^{-1}$ is a diagonal matrix, $\gamma M + \nu_\star I$ can be diagonalized by $Q$. In other words, for $\forall M \in \mathbb{S}_N^+$ in problem (15), $M$ will share eigenspace with the corresponding optimal solution $G_{\star,C}$. Next, denote $(Z_\star, C_\star)$ as the optimal solution of problem (4), $\mathcal{C} \coloneqq \{Z \mid \|Z\|_F^2 = N\}$ as the feasible set and $f(\cdot, \cdot)$ as the objective function. Since that $Z_\star = \arg\min_{Z \in \mathcal{C}} f(Z, C_\star)$, otherwise contradicts with the optimality of $(Z_\star, C_\star)$, we conclude that $G_\star$ and $M_\star$ share eigenspace, where $M_\star \coloneqq (I - C_\star)(I - C_\star)^\top$.

$\square$

**Theorem 2.** *Suppose that $G$ and $M$ are aligned in the same eigenspaces and $\gamma < \frac{1}{\lambda_{\max}(M)}\frac{\alpha^2}{\alpha + \min\{\frac{d}{N}, 1\}}$, then we have that: a) $\operatorname{rank}(Z_\star) = \min\{d, N\}$, and b) the singular values $\sigma_{Z_\star}^{(i)} = \sqrt{\frac{1}{\gamma\lambda_M^{(i)} + \nu_\star} - \frac{1}{\alpha}}$ for all $i = 1, \ldots, \min\{d, N\}$, where $Z_\star$ and $\nu_\star$ are the primal optimal solution and dual optimal solution, respectively.*

*Proof.* Since $\|\boldsymbol{Z} - \boldsymbol{Z}\boldsymbol{C}\|_F^2 = \mathrm{Tr}\left(\boldsymbol{Z}^\top \boldsymbol{Z}\left(\boldsymbol{I} - \boldsymbol{C}\right)\left(\boldsymbol{I} - \boldsymbol{C}\right)^\top\right)$ and $\|\boldsymbol{Z}\|_F^2 = \mathrm{Tr}(\boldsymbol{Z}^\top \boldsymbol{Z})$, problem (5) is equivalent to:

$$
\begin{aligned}
\min_{\boldsymbol{G}} \quad & -\frac{1}{2}\log\det\left(\boldsymbol{I} + \alpha\boldsymbol{G}\right) + \frac{\gamma}{2}\mathrm{Tr}(\boldsymbol{G}\boldsymbol{M}) \\
\text{s.t.} \quad & \mathrm{Tr}(\boldsymbol{G}) = N, \boldsymbol{G} \succeq \boldsymbol{0},
\end{aligned}
\tag{24}
$$

where $\boldsymbol{G} \coloneqq \boldsymbol{Z}^\top \boldsymbol{Z}$ and $\boldsymbol{M} \coloneqq (\boldsymbol{I} - \boldsymbol{C})(\boldsymbol{I} - \boldsymbol{C})^\top$.

Since that $\boldsymbol{G}$ and $\boldsymbol{M}$ have eigenspaces aligned, we can have $\boldsymbol{G}$ and $\boldsymbol{M}$ diagonalized simultaneously by an orthogonal matrix $\boldsymbol{U}$, i.e., $\boldsymbol{G} = \boldsymbol{U}\boldsymbol{\Lambda}_{\boldsymbol{G}}\boldsymbol{U}^\top, \boldsymbol{M} = \boldsymbol{U}\boldsymbol{\Lambda}_{\boldsymbol{M}}\boldsymbol{U}^\top$. Therefore, problem (24) can be transformed into the eigenvalue optimization problem as follows:

$$
\begin{aligned}
\min_{\{\lambda_{\boldsymbol{G}}^{(i)}\}_{i=1}^{\min\{d,N\}}} \quad & -\frac{1}{2}\sum_{i=1}^{\min\{d,N\}}\log(1 + \alpha\lambda_{\boldsymbol{G}}^{(i)}) + \frac{\gamma}{2}\lambda_{\boldsymbol{M}}^{(i)}\lambda_{\boldsymbol{G}}^{(i)} \\
\text{s.t.} \quad & \sum_{i=1}^{\min\{d,N\}}\lambda_{\boldsymbol{G}}^{(i)} = N, \quad \lambda_{\boldsymbol{G}}^{(i)} \geq 0, \quad \text{for all } i = 1,\dots,\min\{d,N\},
\end{aligned}
\tag{25}
$$

where $\{\lambda_{\boldsymbol{M}}^{(1)},\cdots,\lambda_{\boldsymbol{M}}^{(\min\{d,N\})}\}$ are the diagonal entries of $\boldsymbol{\Lambda}_{\boldsymbol{M}}$ and $\{\lambda_{\boldsymbol{G}}^{(1)},\cdots,\lambda_{\boldsymbol{G}}^{(\min\{d,N\})}\}$ are the diagonal entries of $\boldsymbol{\Lambda}_{\boldsymbol{G}}$. Surprisingly, problem (25) is a convex optimization problem. Thus, the KKT condition is sufficient and necessary to guarantee for the global minimizer.

The Lagrangian of problem (25) is:

$$
\mathcal{L}\left(\{\lambda_{\boldsymbol{G}}^{(i)}\}_{i=1}^{\min\{d,N\}}, \{\mu_i\}_{i=1}^{\min\{d,N\}}, \nu\right) \coloneqq
$$

$$
-\frac{1}{2}\sum_{i=1}^{\min\{d,N\}}\log(1 + \alpha\lambda_{\boldsymbol{G}}^{(i)}) + \frac{\gamma}{2}\lambda_{\boldsymbol{M}}^{(i)}\lambda_{\boldsymbol{G}}^{(i)} - \mu_i\lambda_{\boldsymbol{G}}^{(i)} + \frac{\nu}{2}\Big(\sum_{i=1}^{\min\{d,N\}}\lambda_{\boldsymbol{G}}^{(i)} - N\Big),
\tag{26}
$$

where $\mu_i \geq 0, i = 1,\dots,\min\{d,N\}$ and $\nu$ are the Lagrangian multipliers. The KKT conditions are as follows:

$$
\begin{cases}
\nabla_{\lambda_{\boldsymbol{G}_\star}^{(i)}}\mathcal{L} = 0, & \forall i = 1,\dots,\min\{d,N\}, \tag{27} \\[2mm]
\lambda_{\boldsymbol{G}_\star}^{(i)} \geq 0, & \forall i = 1,\dots,\min\{d,N\}, \tag{28} \\[2mm]
\displaystyle\sum_{i=1}^{\min\{d,N\}}\lambda_{\boldsymbol{G}_\star}^{(i)} = N, & \tag{29} \\[2mm]
\mu_{i\star} \geq 0, & \forall i = 1,\dots,\min\{d,N\}, \tag{30} \\[2mm]
\mu_{i\star}\lambda_{\boldsymbol{G}_\star}^{(i)} = 0, & \forall i = 1,\dots,\min\{d,N\}. \tag{31}
\end{cases}
$$

Then, the stationary condition in (27) is equivalent to:

$$
\mu_{i\star} = \frac{1}{2}\Big(\nu_\star + \gamma\lambda_{\boldsymbol{M}}^{(i)} - \frac{\alpha}{1 + \alpha\lambda_{\boldsymbol{G}_\star}^{(i)}}\Big).
\tag{32}
$$

By using Eqs. (28) and (30)-(32), we come up with the following two cases:

$$
\begin{cases}
\mu_{i\star} > 0 \Rightarrow \lambda_{\boldsymbol{G}_\star}^{(i)} = 0, \ \dfrac{1}{\nu_\star + \gamma\lambda_{\boldsymbol{M}}^{(i)}} - \dfrac{1}{\alpha} < 0, \tag{33} \\[4mm]
\mu_{i\star} = 0 \Rightarrow \lambda_{\boldsymbol{G}_\star}^{(i)} > 0, \lambda_{\boldsymbol{G}_\star}^{(i)} = \dfrac{1}{\nu_\star + \gamma\lambda_{\boldsymbol{M}}^{(i)}} - \dfrac{1}{\alpha} > 0. \tag{34}
\end{cases}
$$

From the above two cases, we conclude that:

$$
\lambda_{\boldsymbol{G}_\star}^{(i)} = \max\Big\{0, \frac{1}{\nu_\star + \gamma\lambda_{\boldsymbol{M}}^{(i)}} - \frac{1}{\alpha}\Big\},
\tag{35}
$$

where $\nu_\star$ satisfies:

$$\sum_{i=1}^{\min\{d,N\}} \max\left\{0, \frac{1}{\nu_\star + \gamma\lambda_M^{(i)}} - \frac{1}{\alpha}\right\} = N. \tag{36}$$

Given that $\gamma < (\alpha - \nu_\star)/\lambda_{\max}(M)$, we have $\frac{1}{\nu_\star + \gamma\lambda_M^{(i)}} - \frac{1}{\alpha} > 0$ for all $i = 1, \ldots, \min\{d, N\}$. Therefore, for the optimal solution $Z_\star$ of problem (5), we conclude that: $\text{rank}(Z_\star) = \min\{d, N\}$ and the singular values $\sigma_{Z_\star}^{(i)} = \sqrt{\frac{1}{\gamma\lambda_M^{(i)} + \nu_\star} - \frac{1}{\alpha}}$, for all $i = 1, \ldots, \min\{d, N\}$.

Note that, the results we established just above rely on a condition $\gamma\lambda_{\max}(M) < \alpha - \nu_\star$ where the $\nu_\star$ is the optimal Lagrangian multiplier, which is set as a fixed value related to $\alpha, \gamma$ and $\lambda_{\max}(M)$. Next, we will develop an upper bound for $\nu_\star$ and justify the fact that we ensure $\nu_\star$ to satisfy the condition $\gamma\lambda_{\max}(M) < \alpha - \nu_\star$ by only adjusting the hyper-parameters $\alpha$ and $\gamma$.

In Eq. (36), we can easily find an upper bound of $\nu_\star$ as:

$$N = \sum_{i=1}^{\min\{d,N\}} \max\left\{0, \frac{1}{\nu_\star + \gamma\lambda_M^{(i)}} - \frac{1}{\alpha}\right\} \leq \frac{\min\{d, N\}}{\nu_\star + \gamma\lambda_{\min}(M)} - \frac{\min\{d, N\}}{\alpha}, \tag{37}$$

$$\Rightarrow \nu_\star \leq \frac{1}{\frac{N}{\min\{d,N\}} + \frac{1}{\alpha}} - \gamma\lambda_{\min}(M), \tag{38}$$

Therefore, we can find a tighten bound between $\alpha - \nu_\star$ and $\gamma\lambda_{\max}(M)$ as:

$$\gamma\lambda_{\max}(M) < \frac{\alpha^2}{\alpha + \min\left\{\frac{d}{N}, 1\right\}} < \frac{\alpha^2}{\alpha + \min\left\{\frac{d}{N}, 1\right\}} + \gamma\lambda_{\min}(M) \leq \alpha - \nu_\star, \tag{39}$$

which means that the condition of $\gamma\lambda_{\max}(M) < \alpha - \nu_\star$ can be reformed as:

$$\gamma < \frac{1}{\lambda_{\max}(M)} \frac{\alpha^2}{\alpha + \min\left\{\frac{d}{N}, 1\right\}} \tag{40}$$

$\square$

**Remark 3.** We notice that (25) is a reverse water-filling problem, where the water level is controlled by $1/\alpha$, as shown in Figure A.1. When $G$ and $M$ have eigenspaces aligned and $\gamma < (\alpha - \nu_\star)/\lambda_{\max}(M)$, we have $\text{rank}(Z_\star) = \min\{d, N\}$ and $\lambda_{G_\star}^{(i)} \neq 0$ for all $i \leq \min\{d, N\}$. When $\gamma \geq (\alpha - \nu_\star)/\lambda_{\max}(M)$, non-zero $\lambda_G^{(i)}$ first disappears for the larger $\lambda_M^{(i)}$.

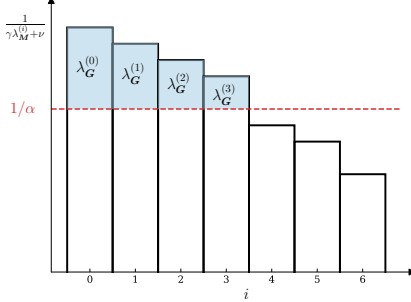

Figure A.1: **Illustration of the optimal solution for problem (25).** The primal problem can be transformed into a classical reverse water-filling problem.

**Theorem 3.** *Suppose that the sufficient conditions to prevent catastrophic feature collapse are satisfied. Without loss of generality, we further assume that the columns in matrix $Z$ are arranged into $k$ blocks according to a certain $N \times N$ permutation matrix $\Gamma$, i.e., $Z = [Z_1, Z_2, \cdots, Z_k]$. Then the*

*condition for that PRO-DSC promotes the optimal solution $(\boldsymbol{Z}_\star, \boldsymbol{C}_\star)$ to be desired structure (i.e., $\boldsymbol{Z}_\star^\top \boldsymbol{Z}_\star$ and $\boldsymbol{C}_\star$ are block-diagonal), is that $\langle (\boldsymbol{I} - \boldsymbol{C})(\boldsymbol{I} - \boldsymbol{C})^\top, \boldsymbol{G} - \boldsymbol{G}^* \rangle \to 0$, where*

$$\boldsymbol{G}^* := \mathrm{Diag}\left(\boldsymbol{G}_{11}, \boldsymbol{G}_{22}, \cdots, \boldsymbol{G}_{kk}\right) = \begin{bmatrix} \boldsymbol{G}_{11} & & \\ & \ddots & \\ & & \boldsymbol{G}_{kk} \end{bmatrix},$$

*and $\boldsymbol{G}_{jj}$ is the block Gram matrix corresponding to $\boldsymbol{Z}_j$.*

*Proof.* We begin with the analysis to the first two terms of the loss function $\tilde{\mathcal{L}} := \mathcal{L}_1 + \gamma \mathcal{L}_2$, where

$$\mathcal{L}_1 := -\frac{1}{2} \log \det \left(\boldsymbol{I} + \alpha (\boldsymbol{Z}\boldsymbol{\Gamma})^\top (\boldsymbol{Z}\boldsymbol{\Gamma})\right) = -\frac{1}{2} \log \det(\boldsymbol{I} + \alpha \boldsymbol{G}),$$

$$\mathcal{L}_2 := \frac{1}{2} \|\boldsymbol{Z}\boldsymbol{\Gamma} - \boldsymbol{Z}\boldsymbol{\Gamma}\boldsymbol{\Gamma}^\top \boldsymbol{C}\boldsymbol{\Gamma}\|_F^2 = \frac{1}{2} \|\boldsymbol{Z} - \boldsymbol{Z}\boldsymbol{C}\|_F^2 = \frac{1}{2} \mathrm{Tr}\left(\boldsymbol{G}\left(\boldsymbol{I} - \boldsymbol{C}\right)\left(\boldsymbol{I} - \boldsymbol{C}\right)^\top\right),$$

since that $\boldsymbol{\Gamma}^\top \boldsymbol{\Gamma} = \boldsymbol{\Gamma}\boldsymbol{\Gamma}^\top = \boldsymbol{I}$. Thus, we have:

$$\tilde{\mathcal{L}}(\boldsymbol{G}, \boldsymbol{C}) = \frac{\gamma}{2} \mathrm{Tr}\left(\boldsymbol{G}\left(\boldsymbol{I} - \boldsymbol{C}\right)\left(\boldsymbol{I} - \boldsymbol{C}\right)^\top\right) - \frac{1}{2} \log \det(\boldsymbol{I} + \alpha \boldsymbol{G}), \tag{41}$$

which is a convex function with respect to (w.r.t) $\boldsymbol{G}$ and $\boldsymbol{C}$, separately. By the property of convex function w.r.t. $\boldsymbol{C}$, we have:

$$\tilde{\mathcal{L}}(\boldsymbol{G}, \boldsymbol{C}) \geq \tilde{\mathcal{L}}(\boldsymbol{G}^*, \boldsymbol{C}^*) + \left\langle \nabla_{\boldsymbol{C}} \tilde{\mathcal{L}}(\boldsymbol{G}^*, \boldsymbol{C}^*), \boldsymbol{C} - \boldsymbol{C}^* \right\rangle + \left\langle \frac{\gamma}{2} \left(\boldsymbol{I} - \boldsymbol{C}\right)\left(\boldsymbol{I} - \boldsymbol{C}\right)^\top, \boldsymbol{G} - \boldsymbol{G}^* \right\rangle$$

$$= \tilde{\mathcal{L}}(\boldsymbol{G}^*, \boldsymbol{C}^*) + \left\langle -\gamma \boldsymbol{G}^*(\boldsymbol{I} - \boldsymbol{C}^*), \boldsymbol{C} - \boldsymbol{C}^* \right\rangle + \left\langle \frac{\gamma}{2} \left(\boldsymbol{I} - \boldsymbol{C}\right)\left(\boldsymbol{I} - \boldsymbol{C}\right)^\top, \boldsymbol{G} - \boldsymbol{G}^* \right\rangle,$$

where $\boldsymbol{C}^* = \mathrm{Diag}\left(\boldsymbol{C}_{11}, \boldsymbol{C}_{22}, \cdots, \boldsymbol{C}_{kk}\right) = \begin{bmatrix} \boldsymbol{C}_{11} & & \\ & \ddots & \\ & & \boldsymbol{C}_{kk} \end{bmatrix}$ with the blocks associating to the

partition of $\boldsymbol{Z} = [\boldsymbol{Z}_1, \boldsymbol{Z}_2, \cdots, \boldsymbol{Z}_k]$. Since that $\left\langle \boldsymbol{G}^*(\boldsymbol{I} - \boldsymbol{C}^*), \boldsymbol{C} - \boldsymbol{C}^* \right\rangle = 0$ due to the complementary between $\boldsymbol{G}^*$ and $\boldsymbol{I} - \boldsymbol{C}^*$, we have:

$$\tilde{\mathcal{L}}(\boldsymbol{G}, \boldsymbol{C}) \geq \tilde{\mathcal{L}}(\boldsymbol{G}^*, \boldsymbol{C}^*) + \left\langle \frac{\gamma}{2} \left(\boldsymbol{I} - \boldsymbol{C}\right)\left(\boldsymbol{I} - \boldsymbol{C}\right)^\top, \boldsymbol{G} - \boldsymbol{G}^* \right\rangle.$$

It is easy to see that if $\left\langle \left(\boldsymbol{I} - \boldsymbol{C}\right)\left(\boldsymbol{I} - \boldsymbol{C}\right)^\top, \boldsymbol{G} - \boldsymbol{G}^* \right\rangle \to 0$, then we will have:

$$\tilde{\mathcal{L}}(\boldsymbol{G}, \boldsymbol{C}) \geq \tilde{\mathcal{L}}(\boldsymbol{G}^*, \boldsymbol{C}^*), \tag{42}$$

where the equality holds *only when* $\boldsymbol{G} = \boldsymbol{G}^*, \boldsymbol{C} = \boldsymbol{C}^*$. Furthermore, if the regularizer $r(\cdot)$ satisfies the extended block diagonal condition as defined in (Lu et al., 2018), then we have that $r(\boldsymbol{C}) \geq r(\boldsymbol{C}^*)$, where the equality holds if and only if $\boldsymbol{C} = \boldsymbol{C}^*$. Therefore, we have:

$$\mathcal{L}(\boldsymbol{G}, \boldsymbol{C}) = \tilde{\mathcal{L}}(\boldsymbol{G}, \boldsymbol{C}) + \beta \cdot r(\boldsymbol{C}) \geq \tilde{\mathcal{L}}(\boldsymbol{G}^*, \boldsymbol{C}^*) + \beta \cdot r(\boldsymbol{C}^*) = \mathcal{L}(\boldsymbol{G}^*, \boldsymbol{C}^*). \tag{43}$$

Thus we conclude that minimizing the loss function $\mathcal{L}(\boldsymbol{G}, \boldsymbol{C}) = \tilde{\mathcal{L}}(\boldsymbol{G}, \boldsymbol{C}) + \beta \cdot r(\boldsymbol{C})$ promotes the optimal solution $(\boldsymbol{G}_\star, \boldsymbol{C}_\star)$ to have block diagonal structure.

We note that the Gram matrix being block-diagonal, i.e., $\boldsymbol{G}_\star = \boldsymbol{G}^*$, implies that $\boldsymbol{Z}_{\star, j_1}^\top \boldsymbol{Z}_{\star, j_2} = \boldsymbol{0}$ for all $1 \leq j_1 < j_2 \leq k$, which is corresponding to the subspaces associated to the blocks $\boldsymbol{Z}_{\star, j}$'s are orthogonal to each other.

$\square$

# B EXPERIMENTAL SUPPLEMENTARY MATERIAL

## B.1 EXPERIMENTAL DETAILS

### B.1.1 SYNTHETIC DATA

As shown in Figure 5a (top row), data points are generated from two manifolds. The first manifold (colored in purple) is generated by sampling 100 data points from

$$\boldsymbol{x} = \begin{bmatrix} \cos\left(\frac{1}{5}\sin\left(5\varphi\right)\right)\cos\varphi \\ \cos\left(\frac{1}{5}\sin\left(5\varphi\right)\right)\sin\varphi \\ \sin\left(\frac{1}{5}\sin\left(5\varphi\right)\right) \end{bmatrix} + \boldsymbol{\epsilon}, \tag{44}$$

where $\varphi$ is taken uniformly from $[0, 2\pi]$ and $\boldsymbol{\epsilon} \sim \mathcal{N}\left(\boldsymbol{0}, 0.05\boldsymbol{I}_3\right)$ is the additive noise. The second manifold (colored in blue) is generated by sampling 100 data points from a Gaussian distribution $\mathcal{N}\left([0,0,1]^\top, 0.05\boldsymbol{I}_3\right)$. To further test more complicated cases, we generate the second manifold by sampling 50 data points from a Gaussian distribution $\mathcal{N}\left([0,0,1]^\top, 0.05\boldsymbol{I}_3\right)$ and 50 data points from another Gaussian distribution $\mathcal{N}\left([0,0,-1]^\top, 0.05\boldsymbol{I}_3\right)$, as shown in Figure 5a (bottom row).

In PRO-DSC, the learnable mappings $\boldsymbol{h}(\cdot; \boldsymbol{\Psi})$ and $\boldsymbol{f}(\cdot; \boldsymbol{\Theta})$ are implemented with two MLPs with Rectified Linear Units (ReLU) (Nair & Hinton, 2010) as the activation function. The hidden dimension and output dimension of the MLPs are set to 100 and 3, respectively. We train PRO-DSC with batch-size $n_b = 200$, learning rate $\eta = 5 \times 10^{-3}$ for 1000 epochs. We set $\gamma = 0.5, \beta = 1000$, and $\alpha = \frac{3}{0.1 \cdot 200}$.

We use DSCNet (Ji et al., 2017) as the representative of the SEDSC methods. In Figure 5b, we set $\gamma = 1$ for both cases, whereas in Figure 5c, $\gamma$ is set to 5 and 100 for the two cases, respectively. The encoder and decoder of DSCNet are MLPs of two hidden layers, with the hidden dimensions being set to 100 and 3, respectively. We train DSCNet with batch-size $n_b = 200$, learning rate $\eta = 1 \times 10^{-4}$ for 1000 epochs.

### B.1.2 REAL-WORLD DATASETS

**Datasets description.** CIFAR-10 and CIFAR-100 are classic image datasets consisting of 50,000 images for training and 10,000 images for testing. They are split into 10 and 100 classes, respectively. CIFAR-20 shares the same images with CIFAR-100 while taking 20 super-classes as labels. ImageNet-Dogs consists of 19,500 images of 15 different dog species. Tiny-ImageNet consists of 100,000 images from 200 different classes. ImageNet-1k is the superset of the two datasets, containing more than 1,280,000 real world images from 1000 classes. For all the datasets except for ImageNet-Dogs, we train the network to implement PRO-DSC on the train set and test it on the test set to validate the generalization of the learned model. For ImageNet-Dogs dataset which does not have a test set, we train the network to implement PRO-DSC on the train set and report the clustering performance on the training set. For a direct comparison, we conclude the basic information of these datasets in Table B.1.

To leverage the CLIP features for training, the input images are first resized to 224 with respect to the smaller edge, then center-cropped to $224 \times 224$ and fed into the CLIP pre-trained image encoder to obtain fixed features.[11] The subsequent training of PRO-DSC takes the extracted features as input, instead of loading the entire CLIP pre-trained model.

**Network architecture and hyper-parameters.** The learnable mappings $\boldsymbol{h}(\cdot; \boldsymbol{\Psi})$ and $\boldsymbol{f}(\cdot; \boldsymbol{\Theta})$ are two fully-connected layers with the same output dimension $d$. Following (Chu et al., 2024), for the experiments on real-world data, we stack a pre-feature layer before the learnable mappings, which is composed of two fully-connected layers with ReLU and batch-norm (Ioffe & Szegedy, 2015).

We train the network by the SGD optimizer with the learning rate set to $\eta = 10^{-4}$, and the weight decay parameters of $\boldsymbol{f}(\cdot; \boldsymbol{\Theta})$ and $\boldsymbol{h}(\cdot; \boldsymbol{\Psi})$ are set to $10^{-4}$ and $5 \times 10^{-3}$, respectively.

---

[11]We use the ViT L/14 pre-trained model provided by `https://github.com/openai/CLIP` for 768-dimensional features.

Table B.1: **Basic statistical information of datasets.** We summarize the information in terms of the data with both the train and test split, as well as the number of classes involved.

| Datasets | # Train | # Test | # Classes |
|---|---|---|---|
| CIFAR-10 | 50,000 | 10,000 | 10 |
| CIFAR-20 | 50,000 | 10,000 | 20 |
| CIFAR-100 | 50,000 | 10,000 | 100 |
| ImageNet-Dogs | 19,500 | N/A | 15 |
| TinyImageNet | 100,000 | 10,000 | 200 |
| ImageNet | 1,281,167 | 50,000 | 1000 |

Following by (Chu et al., 2024), we warm up training $\boldsymbol{f}(\cdot; \boldsymbol{\Theta})$ by diversifying the features with $\mathcal{L}_1 = -\log \det(\boldsymbol{I} + \alpha \boldsymbol{Z}_{\boldsymbol{\Theta}}^\top \boldsymbol{Z}_{\boldsymbol{\Theta}})$ for a few iterations and share the weights to $\boldsymbol{h}(\cdot; \boldsymbol{\Psi})$. We set $\alpha = \frac{d}{0.1 \cdot n_b}$ for all the experiments. We summarize the hyper-parameters for training the network to implement PRO-DSC in Table B.2.

Table B.2: **Hyper-parameters configuration for training the network to implement PRO-DSC with CLIP pre-trained features.** Here $\eta$ is the learning rate, $d_{pre}$ is the hidden and output dimension of pre-feature layer, $m$ is the output dimension of $\boldsymbol{h}$ and $\boldsymbol{f}$, $n_b$ is the batch size for training, and "# warm-up" is the number of iterations of warm-up stage.

| | $\eta$ | $d_{pre}$ | $d$ | #epochs | $n_b$ | #warm-up | $\gamma$ | $\beta$ |
|---|---|---|---|---|---|---|---|---|
| CIFAR-10 | $1 \times 10^{-4}$ | 4096 | 128 | 10 | 1024 | 200 | $300/n_b$ | 600 |
| CIFAR-20 | $1 \times 10^{-4}$ | 4096 | 256 | 50 | 1500 | 0 | $600/n_b$ | 300 |
| CIFAR-100 | $1 \times 10^{-4}$ | 4096 | 128 | 100 | 1500 | 200 | $150/n_b$ | 500 |
| ImageNet-Dogs | $1 \times 10^{-4}$ | 4096 | 128 | 200 | 1024 | 0 | $300/n_b$ | 400 |
| TinyImageNet | $1 \times 10^{-4}$ | 4096 | 256 | 100 | 1500 | 0 | $200/n_b$ | 400 |
| ImageNet | $1 \times 10^{-4}$ | 4096 | 1024 | 100 | 2048 | 2000 | $800/n_b$ | 400 |
| MNIST | $1 \times 10^{-4}$ | 4096 | 128 | 100 | 1024 | 200 | $700/n_b$ | 400 |
| F-MNIST | $1 \times 10^{-4}$ | 1024 | 128 | 200 | 1024 | 400 | $50/n_b$ | 100 |
| Flowers | $1 \times 10^{-4}$ | 1024 | 256 | 200 | 1024 | 200 | $400/n_b$ | 200 |

**Running other algorithms.** Since that $k$-means (MacQueen, 1967), spectral clustering (Shi & Malik, 2000), EnSC (You et al., 2016a), SSCOMP (You et al., 2016b), and DSCNet (Ji et al., 2017) are based on transductive learning, we evaluate these models directly on the test set for all the experiments.

- For EnSC, we tune the hyper-parameter $\gamma \in \{1, 2, 5, 10, 20, 50, 100, 200, 400, 800, 1600, 3200\}$ and the hyper-parameter $\tau$ in $\tau \| \cdot \|_1 + \frac{1-\tau}{2} \| \cdot \|_2^2$ to balance the $\ell_1$ and $\ell_2$ norms in $\{0.9, 0.95, 1\}$ and report the best clustering result.

- For SSCOMP, we tune the hyper-parameter to control the sparsity $k_{\max} \in \{1, 2, 5, 10, 20, 50, 100, 200\}$ and the residual $\epsilon \in \{10^{-4}, 10^{-5}, 10^{-6}, 10^{-7}\}$ and report the best clustering result.

- To apply DSCNet to the CLIP features, we use MLPs with two hidden layers to replace the convolutional encoder and decoder. The hidden dimension of the MLPs are set to 128. We tune the balancing hyper-parameters $\gamma \in \{1, 2, 3, 4\}$ and $\beta \in \{1, 5, 25, 50, 75, 100\}$ and train the model for 100 epochs with learning rate $\eta = 1 \times 10^{-4}$ and batch-size $n_b$ equivalent to number of samples in the test data set.

- As the performance of CPP is evaluated by averaging the ACC and NMI metrics tested on each batch, we reproduce the results by their open-source implementation and report the results on the entire test set. The authors provide two implementations (see `https://github.com/LeslieTrue/CPP/blob/main/main.py` and `https://github.com/LeslieTrue/CPP/blob/main/main_efficient.py`), where one optimizes the cluster head and the feature head separately and the other shares weights between the two heads. In this paper, we test both cases and report the better results.

- For $k$-means and spectral clustering (including when spectral clustering is used as the final step in subspace clustering), we repeat the clustering 10 times with different random initializations (by setting $n_{init} = 10$ in scikit-learn) and report the best results.
- For SENet, SCAN and EDESC, we adjust the hyper-parameters and repeat experiments for three times, with only the best results are reported.

### B.2 EMPIRICAL VALIDATION ON THEORETICAL RESULTS

**Empirical Validation on Theorem 1.** To validate Theorem 1 empirically, we conduct experiments on CIFAR-100 with the same training configurations as described in Section B.1.2 but change the training period to 1000 epochs. For each epoch, we compute $\boldsymbol{G}_b = \boldsymbol{Z}_b^\top \boldsymbol{Z}_b$ and $\boldsymbol{M}_b = (\boldsymbol{I} - \boldsymbol{C}_b)(\boldsymbol{I} - \boldsymbol{C}_b)^\top$ with the learned representations $\boldsymbol{Z}_b$ and self-expressive matrix $\boldsymbol{C}_b$ in mini-batch of size $n_b$ after the last iteration of different epoch. Then, to quantify the eigenspace alignment of $\boldsymbol{G}_b$ and $\boldsymbol{M}_b$, we directly plot the alignment error which is computed via the Frobenius norm of the commutator $\boldsymbol{L} := \|\boldsymbol{G}_b \boldsymbol{M}_b - \boldsymbol{M}_b \boldsymbol{G}_b\|_F$ in mini-batch of size $n_b$ during the training period in Figure 1a. We also show the standard deviation with shaded region after repeating the experiments for 5 random seeds. As can be read, the alignment error decreases monotonically during the training period, implying that the eigenspaces are progressively aligned. Moreover, we find the eigenvector $\{\boldsymbol{u}_j\}$ of $\boldsymbol{M}_b$ by eigenvalue decomposition, where $\boldsymbol{u}_j$ denotes the $j$-th eigenvector which are sorted according to the eigenvalues in the ascending order, and calculate the normalized correlation coefficient which is defined as $\langle \boldsymbol{u}_j, \boldsymbol{G}_b \boldsymbol{u}_j / \|\boldsymbol{G}_b \boldsymbol{u}_j\|_2 \rangle$. Note that when the eigenspace alignment holds, one can verify that:

$$\langle \boldsymbol{u}_j, \frac{\boldsymbol{G}_b \boldsymbol{u}_j}{\|\boldsymbol{G}_b \boldsymbol{u}_j\|_2} \rangle = \begin{cases} 1, & \lambda_{\boldsymbol{G}_b}^{(j)} \neq 0 \\ 0, & \lambda_{\boldsymbol{G}_b}^{(j)} = 0 \end{cases} \quad \text{for all } j = 1, 2, \ldots, n_b. \tag{45}$$

As shown in Figure 1b, the normalized correlation curves associated to the first $d = 128$ eigenvectors converge to 1, whereas the rest converge to 0, implying the progressively alignment between $\boldsymbol{G}_b$ and $\boldsymbol{M}_b$.

**Empirical Validation on Theorem 2.** To verify Theorem 2, we conduct experiments on CIFAR-10 and CIFAR-100. The experimental setup keeps the same as described in Section B.1.2. In each epoch, we compute $\boldsymbol{G}_b = \boldsymbol{Z}_b^\top \boldsymbol{Z}_b$ and $\boldsymbol{M}_b = (\boldsymbol{I} - \boldsymbol{C}_b)(\boldsymbol{I} - \boldsymbol{C}_b)^\top$ from $\boldsymbol{Z}_b$ and $\boldsymbol{C}_b$ in mini-batch after the last iteration, respectively, and then find the eigenvalues of $\boldsymbol{G}_b$ and $\boldsymbol{M}_b$. We display the eigenvalue curves in Figure 1c and 1d, respectively. To enhance the clarity of the visualization, the eigenvalues of $\boldsymbol{G}_b$ and $\boldsymbol{M}_b$ are sorted in descending and ascending order, respectively. As can be observed, there are $\min\{d, n_b\} = 128$ non-zero eigenvalues in $\boldsymbol{G}_b$, approximately being inversely proportional to the smallest 128 eigenvalues of $\boldsymbol{M}_b$. This results empirically demonstrate that $\operatorname{rank}(\boldsymbol{Z}_\star) = \min\{d, N\}$ and $\lambda_{\boldsymbol{G}_\star}^{(i)} = \frac{1}{\gamma \lambda_{\boldsymbol{M}}^{(i)} + \nu_\star} - \frac{1}{\alpha}$ for minimizers.

Furthermore, to verify the sufficient condition of PRO-DSC to prevent feature space collapse, we conduct experiments on CIFAR-10 and CIFAR-100 with varying $\alpha$ and $\gamma$, keeping all the other hyper-parameters consistent with Table B.2. As can be read in Figure 2, Theorem 2 is verified since that $\gamma < \frac{1}{\lambda_{\max}(\boldsymbol{M}_b)} \frac{\alpha^2}{\alpha + \min\{\frac{d}{N}, 1\}}$ yields satisfactory clustering accuracy (ACC%) and subspace-preserving representation error (SRE%). The satisfactory ACC and SRE confirm that PRO-DSC avoids catastrophic collapse when $\gamma < \frac{1}{\lambda_{\max}(\boldsymbol{M})} \frac{\alpha^2}{\alpha + \min\{\frac{d}{N}, 1\}}$ holds. When $\gamma \geq \frac{1}{\lambda_{\max}(\boldsymbol{M}_b)} \frac{\alpha^2}{\alpha + \min\{\frac{d}{N}, 1\}}$, PRO-DSC yields significantly worse ACC and SRE. There is a phase transition phenomenon that corresponds to the sufficient condition to prevent collapse.[12]

**Empirical Validation on Theorem 3.** To intuitively visualize the structured representations learned by PRO-DSC, we visualize the Gram matrices $|\boldsymbol{Z}^\top \boldsymbol{Z}|$ and Principal Component Analysis (PCA) results for both CLIP features and learned representations on CIFAR-10. The experimental setup also keeps the same as described in Section B.1.2.

The Gram matrix shows the similarities between representations within the same class (indicated by in-block diagonal values) and across different classes (indicated by off-block diagonal values).

---

[12]In experiments, we estimate that $\lambda_{\max}(\boldsymbol{M}_b) = 1$ and thus the condition reduces to $\gamma < \frac{\alpha^2}{\alpha + \min\{\frac{d}{N}, 1\}}$.

Moreover, we display the dimensionality reduction results via PCA for the CLIP features and the learned representation of samples from three categories in CIFAR-10. We use PCA for dimensionality reduction as it performs a linear projection, well preserving the underlying structure.

As can be observed in Figure 3, the CLIP features from three classes approximately lie on different subspaces. Despite of the structured nature of the features, the underlying subspaces are not orthogonal. In the Gram matrix of the CLIP feature, the average similarity between features from different classes is greater than 0.6, resulting in an unclear block diagonal structure. After training with PRO-DSC, the spanned subspaces of the learned representations become orthogonal.[13] Additionally, the off-block diagonal values of the Gram matrix decrease significantly, revealing a clear block diagonal structure. These visualization results qualitatively verify that PRO-DSC aligns the representations with a union of orthogonal subspaces.[14]

### B.3 More Experimental Results

#### B.3.1 More results of PRO-DSC on synthetic data

To explore the learning ability of our PRO-DSC, we prepare experiments on synthetic data with adding an additional subspace, as presented in Figure B.1.

**In case 1**, we sample 100 points from Gaussian distribution $\boldsymbol{x} \sim \mathcal{N}([\frac{1}{\sqrt{2}}, 0, \frac{1}{\sqrt{2}}]^\top, 0.05\boldsymbol{I}_3)$ and 100 points from $\boldsymbol{x} \sim \mathcal{N}([-\frac{1}{\sqrt{2}}, 0, \frac{1}{\sqrt{2}}]^\top, 0.05\boldsymbol{I}_3)$, respectively. We train PRO-DSC with batch-size $n_b = 300$, learning rate $\eta = 5 \times 10^{-3}$ for 5000 epochs and set $\gamma = 1.3, \beta = 500, \alpha = \frac{3}{0.1 \cdot 300}$. We observe that our PRO-DSC successfully eliminates the nonlinearity in representations and maximally separates the different subspaces.

**In case 2**, we add a vertical curve

$$\boldsymbol{x} = \begin{bmatrix} \cos\left(\frac{1}{5}\sin\left(5\varphi\right)\right)\cos\varphi \\ \sin\left(\frac{1}{5}\cos\left(5\varphi\right)\right) \\ \cos\left(\frac{1}{5}\sin\left(5\varphi\right)\right)\sin\varphi \end{bmatrix} + \boldsymbol{\epsilon}, \tag{46}$$

from which 100 points are sampled, where $\boldsymbol{\epsilon} \sim \mathcal{N}(\boldsymbol{0}, 0.05\boldsymbol{I}_3)$. We use $\sin(\frac{1}{5}\cos(5\varphi))$ to avoid overlap in the intersection of the two curves. We train PRO-DSC with batch-size $n_b = 200$, learning rate $\eta = 5 \times 10^{-3}$ for 8000 epochs and set $\gamma = 0.5, \beta = 500, \alpha = \frac{3}{0.1 \cdot 200}$. We observe that PRO-DSC finds difficulties to learn representations of data which are located at the intersections of the subspaces. However, for those data points which are away from the intersections are linearized well.

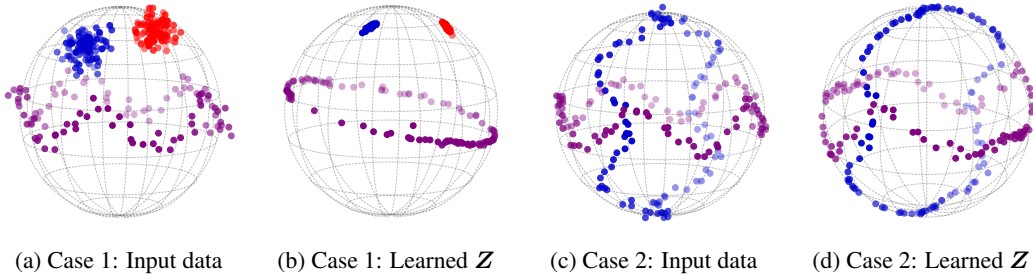

(a) Case 1: Input data     (b) Case 1: Learned $\boldsymbol{Z}$     (c) Case 2: Input data     (d) Case 2: Learned $\boldsymbol{Z}$

Figure B.1: Additional results on synthetic data.

#### B.3.2 Experiments with BYOL Pre-training

To validate the effectiveness of our PRO-DSC without using CLIP features, we conduct a fair comparison with existing deep clustering approaches. Similar to most deep clustering algorithms, we

---

[13]The dimension of each subspace is much greater than one (see Figure B.4). The 1-dimensional subspaces observed in the PCA results are a consequence of dimensionality reduction.

[14]Please refer to Figure B.3 and B.7 for the results on other datasets and the visualization of the bases of each subspace.

divide the training process into two steps. We begin with pre-training the parameters of the backbone with BYOL (Grill et al., 2020). Then, we leverage the parameters pre-trained in the first stage and fine-tune the model by the proposed PRO-DSC loss function. Specifically, we set the learning rate $\eta = 0.05$ and the batch size $n_b = 256$. The output feature dimension $d$ is consistent with the setting for training with the CLIP features. Following (Li et al., 2021; Huang et al., 2023), We use ResNet-18 as the backbone for the experiments on CIFAR-10 and CIFAR-20, and use ResNet-34 as the backbone for the experiments on other datasets, and use a convolution filter of size $3 \times 3$ and stride 1 to replace the first convolution filter. We use the commonly used data augmentations methods to the input images, which are listed as follows:

```
transforms.RandomResizedCrop(size=img_size, scale=(0.08, 1)),
transforms.RandomHorizontalFlip(),
transforms.RandomApply([transforms.ColorJitter(0.4, 0.4, 0.2,
0.1)], p=0.8),
transforms.RandomGrayscale(p=0.2),
transforms.RandomApply([transforms.GaussianBlur(kernel_size=23,
sigma=(0.1, 2.0))], p=1.0).
```

When re-implementing other baselines, we use the code provided by the respective authors and report the best performance after fine-tuning the hyper-parameters.

We report the clustering results based on BYOL pre-training in Table B.3. As can be read from Table B.3, our PRO-DSC outperforms all the deep clustering baselines, including CC (Li et al., 2021), GCC (Zhong et al., 2021), NNM (Dang et al., 2021), SCAN (Van Gansbeke et al., 2020), NMCE (Li et al., 2022), IMC-SwAV (Ntelemis et al., 2022), and MLC (Ding et al., 2023).

Table B.3: **Clustering performance comparison on BYOL pre-training.** The best results are in bold font and the second best results are underlined. Performance marked with "*" is based on our re-implementation.

| Method | CIFAR-10 | | CIFAR-20 | | CIFAR-100 | | TinyImgNet-200 | | ImgNetDogs-15 | |
|---|---|---|---|---|---|---|---|---|---|---|
| | ACC | NMI | ACC | NMI | ACC | NMI | ACC | NMI | ACC | NMI |
| $k$-means | 22.9 | 8.7 | 13.0 | 8.4 | 9.2 | 23.0 | 2.5 | 6.5 | 10.5 | 5.5 |
| SC | 24.7 | 10.3 | 13.6 | 9.0 | 7.0 | 17.0 | 2.2 | 6.3 | 11.1 | 3.8 |
| CC | 79.0 | 70.5 | 42.9 | 43.1 | 26.9* | 48.1* | 14.0 | 34.0 | 42.9 | 44.5 |
| GCC | 85.6 | 76.4 | 47.2 | 47.2 | 28.2* | 49.9* | 13.8 | 34.7 | 52.6 | 49.0 |
| NNM | 84.3 | 74.8 | 47.7 | 48.4 | 41.2 | 55.1 | - | - | 31.1* | 34.3* |
| SCAN | 88.3 | 79.7 | 50.7 | 48.6 | 34.3 | 55.7 | - | - | 29.6* | 30.3* |
| NMCE | 89.1 | 81.2 | 53.1 | 52.4 | 40.0* | 53.9* | 21.6* | 40.0* | 39.8 | 39.3 |
| IMC-SwAV | 89.7 | 81.8 | 51.9 | 52.7 | 45.1 | 67.5 | 28.2 | **52.6** | - | - |
| MLC | 92.2 | 85.5 | **58.3** | 59.6 | 49.4 | **68.3** | 28.7* | 52.2* | 71.0* | 68.3* |
| Our PRO-DSC | **93.0**$_{\pm0.6}$ | **86.5**$_{\pm0.2}$ | **58.3**$_{\pm0.9}$ | **60.1**$_{\pm0.6}$ | **56.3**$_{\pm0.6}$ | 66.7.0$_{\pm1.0}$ | **31.1**$_{\pm0.3}$ | 46.0$_{\pm1.0}$ | **74.1**$_{\pm0.5}$ | **69.5**$_{\pm0.6}$ |

### B.3.3 MORE EXPERIMENTS ON CLIP, DINO AND MAE PRE-TRAINED FEATURES

**Clustering on learned representations.** To quantitatively validate the effectiveness of the structured representations learned by PRO-DSC, we illustrate the clustering accuracy of representations learned by various algorithms in Figure 6. Here, to compared with the representations learned from SEDSC methods, we additionally conduct experiments on DSCNet (Ji et al., 2017) and report the performance in Table B.4. To apply DSCNet on CLIP features, we use MLPs with two hidden layers to replace the stacked convolutional encoder and decoder. As demonstrated in Sec. B.1, we report the best clustering results after the tuning of hyper-parameters. As analyzed in (Haeffele et al., 2021) and Section 2.1, DSCNet overly compresses the representations and yields unsatisfactory clustering results.

**Out of domain datasets.** We evaluate the capability to refine features by training PRO-DSC with pre-trained CLIP features on out-of-domain datasets, namely, MNIST (Deng, 2012), Fashion MNIST (Xiao et al., 2017) and Oxford flowers (Nilsback & Zisserman, 2008). As shown in Table B.5, CPP (Chu et al., 2024) refines the CLIP features and yields better clustering performance comparing with spectral clustering (Shi & Malik, 2000) and EnSC (You et al., 2016a). Our PRO-DSC further demonstrates the best performance on all benchmarks, validating its effectiveness in refining input features.

Table B.4: **Clustering accuracy of CLIP features and learned representations.** We apply $k$-means, spectral clustering, and EnSC to cluster the representations.

| | CIFAR-10 | | | CIFAR-100 | | | CIFAR-20 | | | TinyImgNet-200 | | |
|---|---|---|---|---|---|---|---|---|---|---|---|---|
| | $k$-means | SC | EnSC | $k$-means | SC | EnSC | $k$-means | SC | EnSC | $k$-means | SC | EnSC |
| CLIP | 74.7 | 70.2 | 95.4 | 52.8 | 66.4 | 67.0 | 46.9 | 49.2 | **60.8** | 54.1 | 62.8 | 64.5 |
| SEDSC | 16.4 | 18.9 | 16.9 | 5.4 | 4.9 | 5.3 | 11.7 | 10.6 | 12.8 | 5.7 | 3.9 | 7.2 |
| CPP | 71.3 | 70.3 | **95.6** | 75.3 | 75.0 | 77.5 | 55.5 | 43.6 | 58.3 | 62.1 | 58.0 | 67.0 |
| PRO-DSC | **93.4** | **92.1** | 95.5 | **76.5** | **75.2** | **77.6** | **66.0** | **59.7** | 60.0 | **67.6** | **67.0** | **69.5** |

Table B.5: Experiments on out-of-domain datasets.

| Methods | MNIST | | F-MNIST | | Flowers | |
|---|---|---|---|---|---|---|
| | ACC | NMI | ACC | NMI | ACC | NMI |
| Spectral Clustering (Shi & Malik, 2000) | 74.5 | 67.0 | 64.3 | 56.8 | 85.6 | 94.6 |
| EnSC (You et al., 2016a) | 91.0 | 85.3 | 69.1 | 65.1 | 90.0 | 95.9 |
| CPP (Chu et al., 2024) | 95.7 | 90.4 | 70.9 | 68.8 | 91.3 | 96.4 |
| PRO-DSC | **96.1** | **90.9** | **71.3** | **70.3** | **92.0** | **97.4** |

**Experiments on block diagonal regularizer with different $k$.** To test the robustness of block diagonal regularizer $\|\boldsymbol{A}\|_{\boxed{K}}$ to different $k$, we vary $k$ and report the clustering performance in Table B.6. As illustrated, $k$ does not necessarily equal to the number of clusters. There exists an interval within which the regularizer works effectively.

Table B.6: Clustering performance with different $k$ in block diagonal regularizer.

| | $k$ | 2 | 5 | 10 | 15 | 20 | 25 | 30 |
|---|---|---|---|---|---|---|---|---|
| CIFAR-10 | ACC | 97.2 | 97.2 | **97.4** | 96.3 | 96.3 | 95.4 | 94.0 |
| | NMI | 93.2 | 93.2 | **93.5** | 92.0 | 92.0 | 90.7 | 88.6 |

| | $k$ | 25 | 50 | 75 | 100 | 125 | 150 | 200 |
|---|---|---|---|---|---|---|---|---|
| CIFAR-100 | ACC | 74.3 | 76.7 | 78.1 | 78.2 | **78.9** | 76.4 | 74.8 |
| | NMI | 80.9 | 82.3 | **83.2** | 82.9 | **83.2** | 82.2 | 81.5 |

But if $k$ is significantly smaller than the number of clusters, the effect of block diagonal regularizer will be subtle. Therefore, the performance of PRO-DSC will be similar to that of PRO-DSC without a regularizer (see ablation studies in Section 3). In contrary, if $k$ is significantly larger than the number of clusters, over-segmentation will occur to the affinity matrix, which has negative impact on the subsequent clustering performance.

**Clustering on ImageNet-1k with DINO and MAE.** To test the performance of PRO-DSC based on more pre-trained features other than CLIP (Radford et al., 2021), we further conduct experiments on ImageNet-1k (Deng et al., 2009) pre-trained by DINO (Caron et al., 2021) and MAE (He et al., 2022) (see Table B.7).

DINO and MAE are pre-trained on ImageNet-1k without leveraging external training data, thus their performance on PRO-DSC is lower than CLIP. Similar to the observations in CPP (Chu et al., 2024), DINO initializes PRO-DSC well, yet MAE fails, which is attributed to the fact that features from MAE prefer fine-tuning with labels, while they are less suitable for learning inter-cluster discriminative representations (Oquab et al., 2024). We further extract features from the validation set of ImageNet-1k and visualize through $t$-SNE (Van der Maaten & Hinton, 2008) to validate the hypothesis (see Figure B.2).

### B.3.4 EXPERIMENTS WITHOUT USING PRE-TRAINED MODELS

**Experiments on Reuters and UCI HAR.** During the rebuttal, we conducted extra experiments on datasets Reuters and UCI HAR. The dataset Reuters-10k consists of four text classes, containing 10,000 samples of 2,000 dimension. The UCI HAR is a time-series dataset, consisting of six classes, 10,299 samples of 561 dimension. We take EDESC (Cai et al., 2022) as the baseline method for

Table B.7: Clustering Performance of PRO-DSC based on DINO and CLIP pre-trained features on ImageNet-1k.

| Method | Backbone | PRO-DSC | | $k$-means | |
|---|---|---|---|---|---|
| | | ACC | NMI | ACC | NMI |
| MAE (He et al., 2022) | ViT L/16 | 9.0 | 49.1 | 9.4 | 49.3 |
| DINO (Caron et al., 2021) | ViT B/16 | 57.3 | 79.3 | 52.2 | 79.2 |
| DINO (Caron et al., 2021) | ViT B/8 | 59.7 | 80.8 | **54.6** | **80.5** |
| CLIP (Radford et al., 2021) | ViT L/14 | **65.1** | **83.6** | 52.5 | 79.7 |

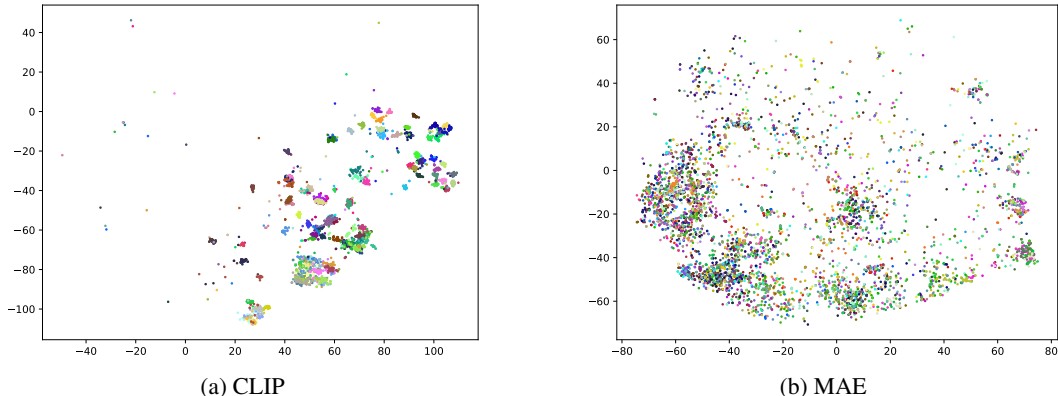

(a) CLIP            (b) MAE

Figure B.2: The $t$-SNE visualization of CLIP and MAE features on the validation set of ImageNet-1k.

deep subspace clustering on Reuters-10k, and take N2D (McConville et al., 2021) and FCMI (Zeng et al., 2023) as the baseline methods for UCI HAR, in which the results are directly cited from the respective papers. We conducted experiments with PRO-DSC on Reuters and UCI HAR following the same protocol for data processing as the baseline methods. We train and test PRO-DSC on the entire dataset and report the results over 10 trials. Experimental results are provided in Table B.8. The hyper-parameters used for PRO-DSC is listed in Table B.9.

Table B.8: Experimental Results on Datasets Reuters and UCI HAR with 10 trials. The results of other methods are cited from the respective papers.

| Dataset | REUTERS-10k | | UCI HAR | |
|---|---|---|---|---|
| | ACC | NMI | ACC | NMI |
| $k$-means (MacQueen, 1967) | 52.4 | 31.2 | 59.9 | 58.8 |
| SC (Shi & Malik, 2000) | 40.2 | 37.5 | 53.8 | 74.1 |
| AE (Bengio et al., 2006) | 59.7 | 32.3 | 66.3 | 60.7 |
| VAE (Kingma & Welling, 2014) | 62.5 | 32.9 | - | - |
| JULE (Yang et al., 2016) | 62.6 | 40.5 | - | - |
| DEC (Xie et al., 2016) | 75.6 | 68.3 | 57.1 | 65.5 |
| DSEC (Chang et al., 2018) | 78.3 | **70.8** | - | - |
| EDESC (Cai et al., 2022) | 82.5 | 61.1 | - | - |
| DFDC (Zhang & Davidson, 2021) | - | - | 86.2 | **84.5** |
| N2D (McConville et al., 2021) | - | - | 82.8 | 71.7 |
| FCMI (Zeng et al., 2023) | - | - | **88.2** | 80.7 |
| PRO-DSC | **85.7** $\pm$ 1.3 | 64.6 $\pm$ 1.3 | 87.1 $\pm$ 0.4 | 80.9 $\pm$ 1.2 |

Table B.9: Configuration of hyper-parameters for experiments on Reuters, UCI HAR, EYale-B, ORL and COIL-100.

| Dataset | $\eta$ | $d_{pre}$ | $d$ | #epochs | $n_b$ | #warm-up | $\gamma$ | $\beta$ |
|---|---|---|---|---|---|---|---|---|
| REUTERS-10k | $10^{-4}$ | 1024 | 128 | 100 | 1024 | 50 | 50 | 200 |
| UCI HAR | $10^{-4}$ | 1024 | 128 | 100 | 2048 | 20 | 100 | 300 |
| EYale-B | $10^{-4}$ | 1080 | 256 | 10000 | 2432 | 100 | 200 | 50 |
| ORL | $10^{-4}$ | 80 | 64 | 5000 | 400 | 100 | 75 | 10 |
| COIL-100 | $10^{-4}$ | 12800 | 100 | 10000 | 7200 | 100 | 200 | 100 |

**Comparison to AGCSC and SAGSC on Extended Yale B, ORL, and COIL-100.** During the rebuttal, we conducted more experiments on two state-of-the-art subspace clustering methods AGCSC (Wei et al., 2023) and ARSSC (Wang et al., 2023a). Since that both of the two methods cannot handle the datasets used for evaluating our PRO-DSC, we conducted experiments on the datasets: Extended Yale B (EYaleB), ORL, and COIL-100. We set the architecture of pre-feature layer in PRO-DSC as the same to the encoder of DSCNet (Ji et al., 2017). The hyper-parameters configuration for training PRO-DSC is summarized in Table B.9. We repeated experiments for 10 trails and report the average with standard deviation in Table B.10. As baseline methods, we use EnSC (You et al., 2016a), SSCOMP (You et al., 2016b), S$^3$COMP (Chen et al., 2020b), DSCNet, DSSC (Lim et al., 2020) and DELVE Zhao et al. (2024). The results of these methods, except for S$^3$COMP and DELVE, are directly cited them from DSSC (Lim et al., 2020), and the results of S$^3$COMP and DELVE are cited from their own papers.

- Comparison to AGCSC. Our method surpasses AGCSC on the Extended Yale B dataset and achieves comparable results on the ORL dataset. However, AGCSC cannot yield the result on COIL-100 in 24 hours.
- Comparison to ARSSC. ARSSC employs three different non-convex regularizers: $\ell_\gamma$ norm Penalty (LP), Log-Sum Penalty (LSP), and Minimax Concave Penalty (MCP). While ARSSC-MCP performs the best on Extended Yale B, our PRO-DSC outperforms ARSSC-MCP on ORL. While AGCSC performs the best on ORL, but it yields inferior results on Extended Yale B and it cannot yield the results on COIL-100 in 24 hours. Thus, we did not report the results of AGCSC on COIL-100 and marked it as Out of Time (OOT). Our PRO-DSC performs the second best results on Extended Yale B, ORL and the best results on COIL-100. Since that we have not found the open-source code for ARSSC, we are unable to have their results on COIL-100. This comparison also confirms the scalablity of our PRO-DSC which is due to the re-parametrization (similar to SENet).

Table B.10: Experiments on Extended Yale B, ORL and COIL-100.

| | EYale-B | | ORL | | COIL-100 | |
|---|---|---|---|---|---|---|
| | ACC | NMI | ACC | NMI | ACC | NMI |
| EnSC | 65.2 | 73.4 | 77.4 | 90.3 | 68.0 | 90.1 |
| SSCOMP | 78.0 | 84.4 | 66.4 | 83.2 | 31.3 | 58.8 |
| S$^3$COMP-C (Chen et al., 2020b) | 87.4 | - | - | - | 78.9 | - |
| DSCNet | 69.1 | 74.6 | 75.8 | 87.8 | 49.3 | 75.2 |
| DELVE (Zhao et al., 2024) | 89.8 | 90.1 | - | - | 79.0 | 93.9 |
| J-DSSC (Lim et al., 2020) | 92.4 | 95.2 | 78.5 | 90.6 | 79.6 | 94.3 |
| A-DSSC (Lim et al., 2020) | 91.7 | 94.7 | 79.0 | 91.0 | 82.4 | 94.6 |
| AGCSC (Wei et al., 2023) | 92.3 | 94.0 | **86.3** | **92.8** | OOT | OOT |
| ARSSC-LP (Wang et al., 2023a) | 95.7 | - | 75.5 | - | - | - |
| ARSSC-LSP (Wang et al., 2023a) | 95.9 | - | 71.3 | - | - | - |
| ARSSC-MCP (Wang et al., 2023a) | **99.3** | - | 72.0 | - | - | - |
| PRO-DSC | 96.0 ± 0.3 | **95.7** ± 0.8 | 83.2 ± 2.2 | 92.7 ± 0.6 | **82.8** ± 0.9 | **95.0** ± 0.6 |

## B.4 MORE VISUALIZATION RESULTS

**Gram matrices and PCA visualizations.** To qualitatively validate that PRO-DSC learns representations aligning with a union-of-orthogonal-subspaces distribution, we visualize the Gram matrices

and PCA dimension reduction results of CLIP features and learned representations from PRO-DSC for each dataset. As shown in Figure B.3, the off-block diagonal values decrease significantly, implying the orthogonality between representations from different classes. The orthogonal between subspaces can also be observed from the PCA dimension reduction results.

**Singular values visualization.** To show the intrinsic dimension of CLIP features and the representations of PRO-DSC, We plot the singular values of CLIP features and PRO-DSC's representations in Figure B.4. Specifically, the singular values of features from all the samples are illustrated on the left and the singular values of features within each class are illustrated on the middle and right. As can be seen, the singular values of PRO-DSC decrease much slower than that of CLIP, implying that the features of PRO-DSC enjoy a higher intrinsic dimension and more isotropic structure in the ambient space.

**Learning curves.** We plot the learning curves with respect to loss values and performance of PRO-DSC on CIFAR-100, CIFAR-20 and ImageNet-1k in Figure B.5a, Figure B.5b and Figure B.5c, respectively. Recall that $\mathcal{L}_1 := -\frac{1}{2}\log\det\left(\boldsymbol{I} + \alpha\boldsymbol{Z}_{\boldsymbol{\Theta}}^{\top}\boldsymbol{Z}_{\boldsymbol{\Theta}}\right)$, $\mathcal{L}_2 := \frac{1}{2}\|\boldsymbol{Z}_{\boldsymbol{\Theta}} - \boldsymbol{Z}_{\boldsymbol{\Theta}}\boldsymbol{C}_{\boldsymbol{\Psi}}\|_F^2$, and $\mathcal{L}_3 := \|\boldsymbol{A}_{\boldsymbol{\Psi}}\|_{\boxed{K}}$. Since $\mathcal{L}_1$ is the only loss function used in the warm-up stage, we plot all the curves starting from the iteration when warm-up ends. As illustrated, the clustering performance of PRO-DSC steadily increase as the loss values gradually decrease, which shows the effectiveness of the proposed loss functions in PRO-DSC.

$t$-**SNE visualization of learned representations.** We visualize the CLIP features and cluster representations learned by PRO-DSC leveraging $t$-SNE (Van der Maaten & Hinton, 2008) in Figure B.6. As illustrated, the learned cluster representations are significantly more compact compared with the CLIP features, which contributes to the improved clustering performance.

**Subspace visualization.** We visualize the principal components of subspaces learned by PRO-DSC in Figure B.7. For each cluster in the dataset, we apply Principal Component Analysis (PCA) to the learned representations. We select the top eight principal components to represent the learned subspaces. Then, for each principal component, we display eight images whose representations are most closely aligned with that principal component.

Interestingly, we can observe specific semantic meanings from the principal components learned by PRO-DSC. For instance, the third row of Figure B.7a consists of stealth fighters, whereas the fifth row shows airliners. The second row of Figure B.7c consists of birds standing and resting, while the sixth row shows flying eagles. While Figure B.7j consists of all kinds of trucks, the first row shows fire trucks.

## C LIMITATIONS AND FAILURE CASES

**Limitations:** In this paper, we explore an effective framework for deep subspace clustering with theoretical justification. However, it is not clear how to develop the geometric guarantee for our PRO-DSC framework to yield correctly subspace-preserving solution. Moreover, it is an unsupervised learning framework, we left the extension to semi-supervised setting as future work.

**Failure Cases:** In this paper, we evaluate our PRO-DSC framework on four scenarios of synthetic data (Fig. 5 and B.1), six benchmark datasets with CLIP features (Table 1), five benchmark datasets with BYOL pre-trained features (Table B.3), three out-of-domain datasets (Table B.5), using four different regularization terms (Table 3), using different feature extractor (Table B.7) and varying hyper-parameters (Fig. 7 and Table B.6). We also conduct experiments on two face image datasets (Table B.10), text and temporal dataset (Table B.8) . However, as demonstrated in Fig. 1, our PRO-DSC will fail if the sufficient condition to prevent catastrophic collapse is not satisfied by using improper hyper-parameters $\gamma$ and $\alpha$.

**Extensibility:** As a general framework for self-expressive model based deep subspace clustering, our PRO-DSC is reasonable, scalable and flexible to miscellaneous extensions. For example, rather than using $\log\det(\cdot)$, there are other methods to solve the feature collapse issue, e.g., the nuclear norm. In addition, it is also worthwhile to incorporate the supervision information from the pseudo-label, e.g., (Huang et al., 2023; Jia et al., 2025; Li et al., 2017), for further improving the performance of our PRO-DSC.

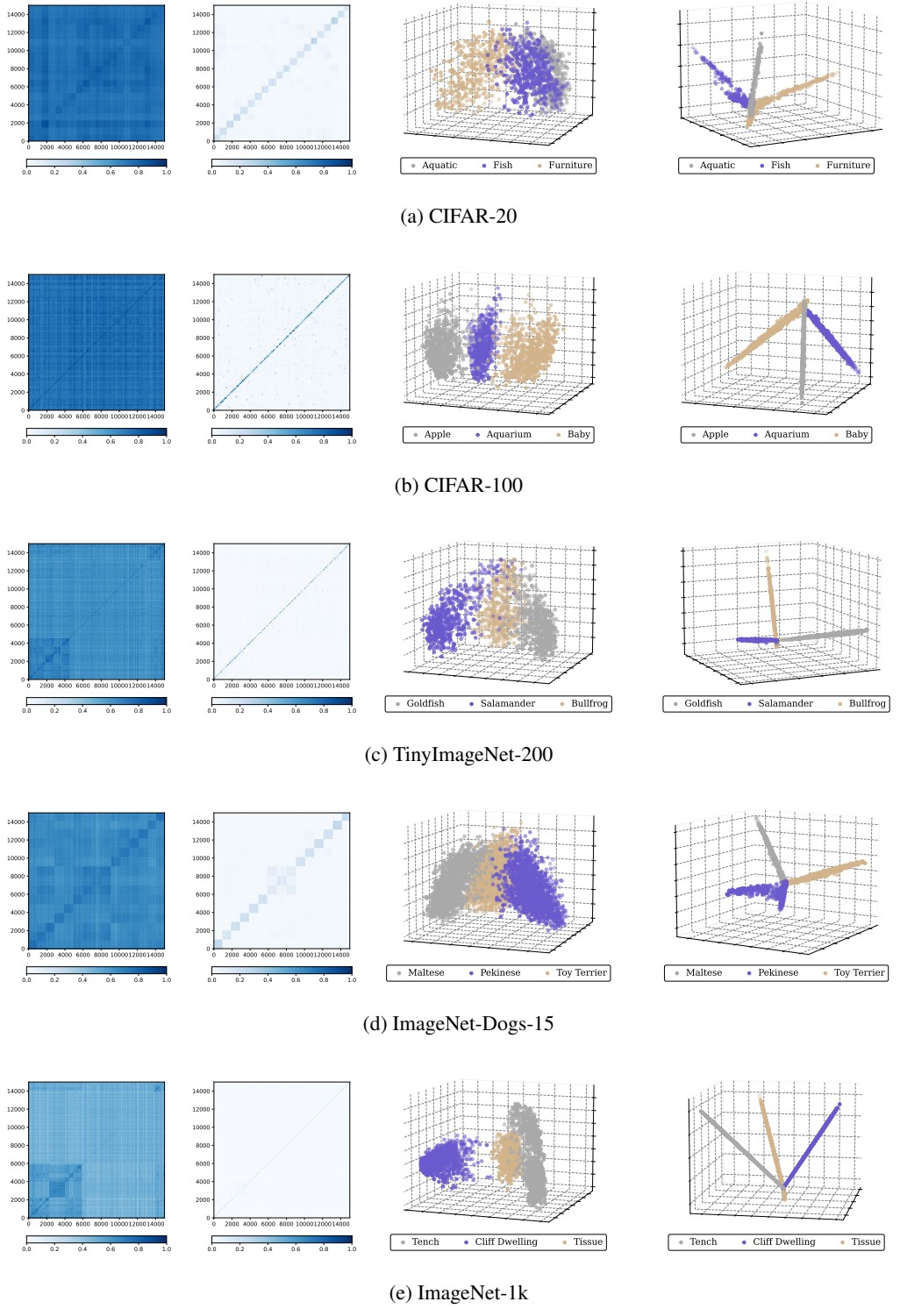

Figure B.3: **Visualization of the union-of-orthogonal-subspaces structure of the learned representations via Gram matrix and PCA dimension reduction on three categories.** Left: $|\boldsymbol{X}^\top \boldsymbol{X}|$. Mid-left: $|\boldsymbol{Z}^\top \boldsymbol{Z}|$. Mid-right: $\boldsymbol{X}_{(3)}$ via PCA. Right: $\boldsymbol{Z}_{(3)}$ via PCA.

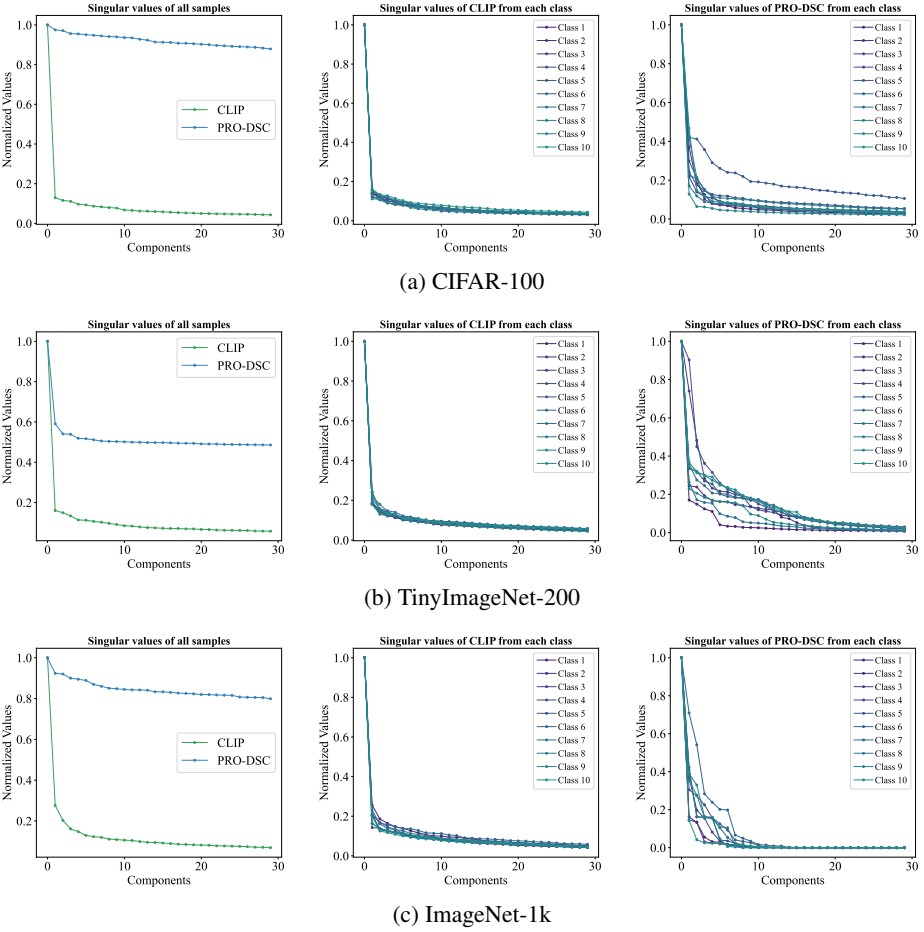

Figure B.4: **Singular values of features from all samples (left) and features from each class (mid and right).** For the better clarity, we plot the singular values for the first ten classes.

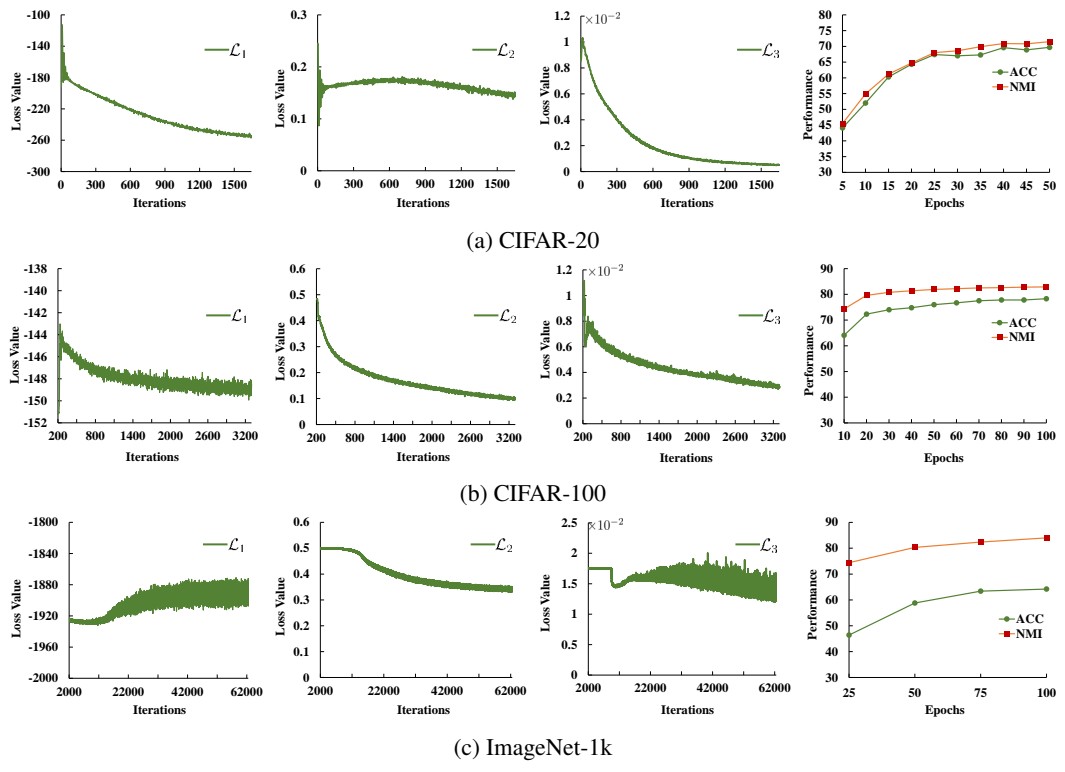

Figure B.5: The learning curves w.r.t. loss values and evaluation performance of PRO-DSC on CIFAR-20, CIFAR-100 and ImageNet-1k dataset.

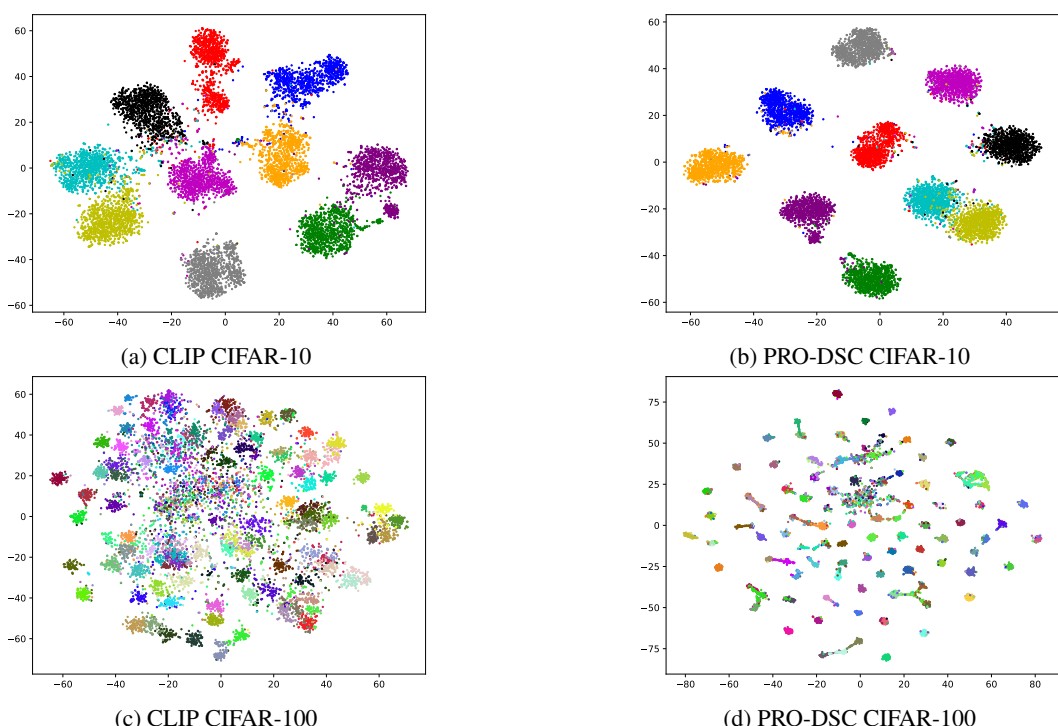

Figure B.6: $t$-**SNE visualization of CLIP features and PRO-DSC's learned representations.** The experiments are conducted on CIFAR-10 and CIFAR-100 dataset.

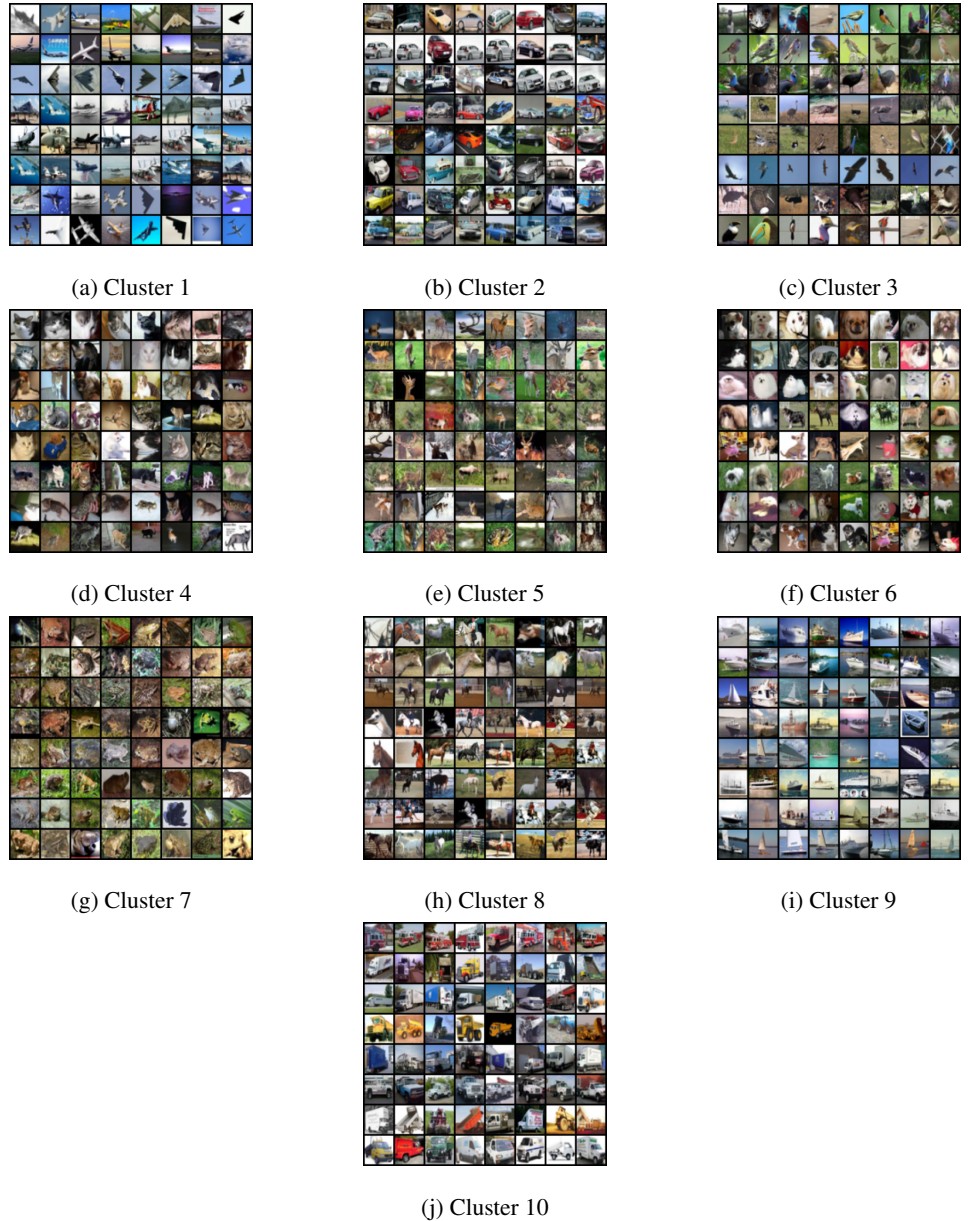

(a) Cluster 1      (b) Cluster 2      (c) Cluster 3

(d) Cluster 4      (e) Cluster 5      (f) Cluster 6

(g) Cluster 7      (h) Cluster 8      (i) Cluster 9

(j) Cluster 10

Figure B.7: **Visualization of the principal components in CIFAR-10 dataset.** For each cluster, we display the most similar images to its principal components.

