# OpenReview forum: "Exploring a Principled Framework for Deep Subspace Clustering"
_ICLR.cc/2025/Conference — ICLR 2025 Poster_

### Official Review · Reviewer_JbyQ · 2024-10-29

**Soundness:** 3
**Presentation:** 3
**Contribution:** 3
**Rating:** 6
**Confidence:** 3

**Summary:**

Deep subspace clustering methods usually encounter the challenge of feature collapse and lack theoretical guarantees for learning representations that form a union of subspaces (UoS) structure. To address these issues, this paper presents a principled framework for deep subspace clustering (PRO-DSC). The framework incorporates an effective regularization on the learned representations to prevent feature space collapse. Furthermore, theoretical analysis demonstrates that PRO-DSC can yield representations of a UoS structure under certain conditions. Experimental results show the effectiveness of the proposed method.

**Strengths:**

1. This work presents an effective method to prevent feature collapse and addresses the lack of theoretical guidance for learning representations with a UoS structure. An efficient implementation is presented to alleviate the computational burden of self-expressive learning.
2. The work is sound from a technical perspective. It combines solid theoretical proof and empirical evidence.
3. Experiments on synthetic and real-world data are conducted to evaluate the effectiveness. Meanwhile, both quantitative and qualitative results are provided for comparison.
4. Overall, this paper is well-written and thoughtfully structured.

**Weaknesses:**

1. As one of the main challenges this work focuses on is learning UoS representations, I suggest the authors emphasize the significance of this challenge by clarifying the associated consequences and providing empirical observations to support it.
2. Since the proposed PRO-DSC is designed as a framework, I expect further discussion and experimentation on its scalability and extensibility.
3. I have several questions and concerns regarding the experimental parts:
    1) Why is only the proposed method run multiple times for evaluation, while other methods seem to be tested only once? Additionally, why is the method run only three times? Repeating the evaluation 10 times is commonly preferred for more reliable results.
    2) Why are only image datasets used for evaluation? I suggest testing the performance on a wider range of datasets, such as the Reuters and UCI HAR datasets.
    3) Most of the comparison methods used in experiments are outdated. Please consider adding two more state-of-the-art subspace clustering methods for comparison, such as AGCSC [1] and SAGSC [2].
    4) Why do the comparison methods differ between experiments in Tables 1 and 2?
    5) In the synthetic data experiments, why is DSCNet used as the representative SEDSC method rather than the more competitive, recent SENET?

[1] Wei, Lai, et al. "Adaptive graph convolutional subspace clustering." Proceedings of the IEEE/CVF Conference on Computer Vision and Pattern Recognition. 2023.

[2] Wang, Libin, et al. "Attention reweighted sparse subspace clustering." Pattern Recognition, 139 (2023): 109438.

**Questions:**

I have several questions and concerns regarding the experimental parts:
1) Why is only the proposed method run multiple times for evaluation, while other methods seem to be tested only once? Additionally, why is the method run only three times? Repeating the evaluation 10 times is commonly preferred for more reliable results.
2) Why are only image datasets used for evaluation? I suggest testing the performance on a wider range of datasets, such as the Reuters and UCI HAR datasets.
3) Most of the comparison methods used in experiments are outdated. Please consider adding two more state-of-the-art subspace clustering methods for comparison, such as AGCSC [1] and SAGSC [2].
4) Why do the comparison methods differ between experiments in Tables 1 and 2?
5) In the synthetic data experiments, why is DSCNet used as the representative SEDSC method rather than the more competitive, recent SENET?

[1] Wei, Lai, et al. "Adaptive graph convolutional subspace clustering." Proceedings of the IEEE/CVF Conference on Computer Vision and Pattern Recognition. 2023.

[2] Wang, Libin, et al. "Attention reweighted sparse subspace clustering." Pattern Recognition, 139 (2023): 109438.

---

> ### Author Response · Authors · 2024-11-21
> **1. Regarding to the Evaluation**
>
> First of all, we would like to express our appreciation to the reviewer for recognizing our theoretical justification, empirical evaluations and writing, and for the constructive comments and insightful suggestions.
>
> It is not the case that other algorithms are tested only one trial.
> In Appendix B.1, we provided the running details of EnSC, SSCOMP, DSCNet, and CPP.
> - For SENet, SCAN and EDESC, we adjust the hyper-parameters and repeat experiments for three times, with only the best results are reported.
> - For TEMI, we directly cited the results from the paper.
> - For our PRO-DSC, we repeated experiments for three trials and report the average results (with standard deviation).
>
> Nevertheless, the two classical methods (i.e., k-means and spectral clustering) are overlooked to report the running details:
> - For k-means and spectral clustering (including when spectral clustering is used as the final step in subspace clustering), we repeat the clustering 10 times with different random initializations (by setting n$\\_$init=10 in scikit-learn) and report the best results.
>
> During the rebuttal, we conducted more experiments on the CLIP features and reported the average results with standard deviation over 10 trials in the revised version. To have a clear comparison, we list the results over 3 trials and 10 trials in Table B.3.  For the experiments that trained from scratch, we will update them in the final version due to time limitation in the rebuttal period.

---

> ### Author Response · Authors · 2024-11-21
> **2. Experimental Results on Datasets Reuters and UCI HAR**
>
> The dataset Reuters-10k consists of four text classes, containing 10,000 samples of 2,000 dimension. The UCI HAR is a time-series dataset, consisting of six classes, 10,299 samples of 561 dimension. We take EDESC as the baseline method for deep subspace clustering on Reuters-10k, and take N2D [1] and FCMI [2] as the baseline methods for UCI HAR, in which the results are directly cited from the respective papers. We conducted experiments with PRO-DSC on Reuters and UCI HAR following the same protocol for data processing as the baseline methods. We train and test PRO-DSC on the entire dataset and report the results over 10 trials.
>
> Experimental results (ACC% / NMI%) are provided in Table B.5.  We observed that our PRO-DSC yields better results than EDESC on REUTERS-10k; but yields competitive results on UCI HAR. Note that our PRO-DSC with a vanilla self-expressive model has already demonstrated highly competitive performance. By incorporating the pseudo-label information, more sophisticated self-supervision strategy or more calibrated self-expressive model might enable PRO-DSC to yield more promising performance. We have added the results in the Table 4 of Appendix B.
>
> | Dataset | REUTERS-10k                 | UCI HAR                 |
> | :------- | :---------------------------: | :-----------------------: |
> | k-means | 52.4 / 31.2                 | 59.9 / 58.8             |
> | SC      | 40.2 / 37.5                 | 53.8 / 74.1             |
> | AE      | 59.7 / 32.3                 | 66.3 / 60.7             |
> | VAE     | 62.5 / 32.9                 | - / -                   |
> | JULE    | 62.6 / 40.5                 | - / -                   |
> | DEC     | 75.6 / 68.3                 | 57.1 / 65.5             |
> | DSEC    | 78.3 / 70.8                 | - / -                   |
> | EDESC   | 82.5 / 61.1                 | - / -                   |
> | DFDC    | - / -                       | 86.2 / 84.5             |
> | N2D [1]     | - / -                       | 82.8 / 71.7             |
> | FCMI [2]   | - / -                       | **88.2** / 80.7         |
> | PRO-DSC | **85.7**±1.3 / **64.6**±1.3 | 87.1±0.4 / **80.9**±1.2 |
>
> [1] McConville, et al., "N2d: (not too) deep clustering via clustering the local manifold of an autoencoded embedding," 25th International Conference on Pattern Recognition (ICPR), Jan. 2021.
>
> [2] Zeng, et al., "Deep fair clustering via maximizing and minimizing mutual information: Theory, algorithm and metric," Proceedings of the IEEE/CVF Conference on Computer Vision and Pattern Recognition, 2023.

---

> ### Author Response · Authors · 2024-11-21
> **3. Comparison to AGCSC and SAGSC**
>
> We thank the reviewer for pointing out two state-of-the-art subspace clustering methods AGCSC (Wei et al. CVPR'23) and ARSSC (Wang et al. PR'23). Since that both of the two methods cannot handle the datasets used for evaluating our PRO-DSC, we conducted experiments on the benchmark datasets: Extended Yale B (EYaleB), ORL, and COIL-100.
> We set the architecture of pre-feature layer in PRO-DSC as the same to the encoder of DSCNet.
>
> We repeated experiments for 10 trials, reported the average results (ACC% / NMI%) with standard deviation in Table B.5 of Appendix and also listed the results as follows.
>
> | Dataset       | EYale-B                               | ORL                                             | COIL-100                                |
> | ------------- | ------------------------------------- | ----------------------------------------------- | --------------------------------------- |
> | EnSC          | 65.2 / 73.4                           | 77.4 / 90.3                                     | 68.0 / 90.1                             |
> | SSCOMP        | 78.0 /84.4                            | 66.4 / 83.2                                     | 31.3 / 58.8                             |
> | S3COMP        | 87.4 / -                              | - / -                                           | 78.9 / -                                |
> | DSCNet        | 69.1 / 74.6                           | 75.8 / 87.8                                     | 49.3 / 75.2                             |
> | J-DSSC [3]    | 92.4 / $\underline{95.2}$             | 78.5 / 90.6                                     | 79.6 / 94.3                             |
> | A-DSSC [3]    | 91.7 / 94.7                           | 79.0 / 91.0                                     | $\underline{82.4}$ / $\underline{94.6}$ |
> | AGCSC  [4]    | 92.3 / 94.0                           | **86.3** / **92.8**                             | OOT / OOT                               |
> | ARSSC-LP [5]  | 95.7 / -                              | 75.5 / -                                        | - / -                                   |
> | ARSSC-LSP [5] | 95.9 / -                              | 71.3 / -                                        | - / -                                   |
> | ARSSC-MCP [5] | **99.3** / -                          | 72.0 / -                                        | - / -                                   |
> | PRO-DSC       | $\underline{96.0}$±0.3 / **95.7**±0.8 | $\underline{83.2}$±2.2 / $\underline{92.7}$±0.6 | **82.8**±0.9 / **95.0**±0.6             |
>
> The results of AGCSC, ARSSC, and DSSC are directly cited from their papers; the resutls of DSCNet, SSCOMP, S3COMP and EnSC are cited from DSSC [3]. We have added these results in Table B.5 of the revised version of our paper.
> The hyper-parameters configuration for training PRO-DSC is summarized as follow (also in Table B.3 in the revised version).
>
> | Dataset     | $\eta$ | $d_{pre}$ | $d$  | $\\#$ epoch | $n_b$ | $\\#$ warm-up | $\gamma$ | $\beta$ |
> | ----------- | ------ | --------- | ---- | ---------- | ----- | ------------ | -------- | ------- |
> | REUTERS-10k | 1e-4   | 1024      | 128  | 100        | 1024  | 50           | 50       | 200     |
> | UCI HAR     | 1e-4   | 1024      | 128  | 100        | 2048  | 20           | 100      | 300     |
> | EYale-B     | 1e-4   | 1024      | 256  | 10000      | 2432  | 100          | 200      | 50      |
> | ORL         | 1e-4   | 80        | 64   | 5000       | 400   | 100          | 75       | 10      |
> | COIL-100    | 1e-4   | 12800     | 100  | 10000      | 7200  | 100          | 200      | 100     |
>
> - AGCSC. Our method surpasses AGCSC on the Extended Yale B dataset and achieves comparable results on the ORL dataset. However, AGCSC did yield the result on COIL-100 in 24 hours.
>
> - ARSSC. ARSSC employs three different non-convex regularizers: $\ell_\gamma$ norm Penalty (LP), Log-Sum Penalty (LSP), and Minimax Concave Penalty (MCP).
>
> While ARSSC-MCP performs the best on Extended Yale B, our PRO-DSC outperforms ARSSC-MCP on ORL. While AGCSC performs the best on ORL, but it yields inferior results on Extended Yale B and it cannot yield the results on COIL-100 in 24 hours. Thus, we did not report the results of AGCSC on COIL-100 and marked it as Out of Time (OOT). Our PRO-DSC performs the second best results on Extended Yale B, ORL and the best results on COIL-100. Since that we have not found the open-source code for ARSSC, we are unable to have its results on COIL-100.
>
> [3] Lim et al.: "Doubly stochastic subspace clustering", arXiv preprint arXiv:2011.14859 (2020).
>
> [4] Wei et al.: "Adaptive graph convolutional subspace clustering", Proceedings of the IEEE/CVF Conference on Computer Vision and Pattern Recognition, 2023.
>
> [5] Wang et al.: "Attention reweighted sparse subspace clustering", Pattern Recognition, 139 (2023): 109438.

---

> ### Author Response · Authors · 2024-11-21
> **4. About the different baselines in Tables 1 and 2**
>
> we apologize for the confusion. The different baselines in Tables 1 and 2 are actually due to the fact that many of the baselines can only handle either the extracted features or the raw images.
>
> To be specific, the reason to use different baseline algorithms to compare in Tables 1 and 2 mainly due to the types of input data that can be handled by different algorithms. Table 1 shows the clustering performance of using extracted CLIP features, so the baselines here must be applicable to the input of vectors. On contrary, all the deep clustering methods in Table 2 rely on data augmentation which is applied to raw image, and thus cannot be applied to the fixed extracted features.
> To further clarify them, we will group the baselines in Tables 1 and 2 for further explanation.
>
> - **The baselines included in both Tables 1 and 2.**
>   For $k$-means and spectral clustering (SC), which are used in both Tables 1 and 2, we take the extracted CLIP features (for Table 1) and the flattened raw images (for Table 2) as their input.
>   For SCAN, when it is performed with the CLIP features (in Table 1), we omit its augmentation consistency based representation learning loss.
>   Notably, CPP (in Table 1) and MLC (in Table 2) are almost the same, which are both based on the MCR2 principle (Yu et al. NeurIPS'20), except for that MLC additionally promote consistency among different augmentations.
>
> - **The baselines are included in Table 1 but not in Table 2.**
>   The scalable subspace clustering algorithms, e.g., SSCOMP, ENSC, SENet, are not suitable to raw image as inputs. Since that DSCNet is not a scalable algorithm, it is challenging to be run on these datasets.
>   Thus, we only conduct experiments on the CLIP features of the test set on CIFAR and ImageNet for these methods. Since that TEMI is specifically designed for CLIP features, it does not appear in Table 2.
>
> - **The baselines are included in Table 2 but not in Table 1.**
>  The methods, including CC, GCC, NNM, NMCE, IMC-SwaV, which rely on data augmentation for unsupervised representation learning, cannot be proper transferred to the CLIP features.
>
> Therefore, we list these algorithms in Tables 1 and 2 of our submission for best presenting their performance and for fair comparison. To compare with these baseline algorithms in both cases, we extend our PRO-DSC to be able to evaluate clustering performance with both types of input.

---

> ### Author Response · Authors · 2024-11-21
> **5. Why use DSCNet, rather than SENet on synthetic data for comparison?**
>
> We note that while SENet is implemented via a query-key network, it does not belong to deep subspace clustering method because the query-key network is used to learn the self-expressive coefficients, rather than to learn the representation of the input data.
>
> DSCNet (Ji et al., NeurIPS'17) is designed by encapsulating a self-expressive (SE) model (which is called a SE layer) in the ``middle'' of a stacked convolutional auto-encoder network (SCAE) and is trained by using a combination of reconstruction loss and SE loss with $\ell_1$ or $\ell_2$ regulizer. While the framework of DSCNet is clear and remarkable clustering accuracy has been reported on four image datasets (especially on Extended Yale B), there is no clear evidence to demonstrate that SCAE is able to amend the input data to align with a union of subspaces (UoS). Even worse, as theoretically justified by (Haeffele et al. ICLR'21), the optimal solution of DSCNet suffers from a catastrophic feature collapse---which is also revealed empirically in the experiments on synthetic data in our submission.

---

> ### Author Response · Authors · 2024-11-21
> **6. Regarding to Scalablity and Extensibility of Our PRO-DSC**
>
> It is noteworthy that the goal of our work is to provide a reasonable (or a principled) framework for deep subspace clustering based on self-expressive (SE) model. We demonstrate theoretically and empirically that adding a $\log \det(\cdot)$ term into the SE model can prevent the catastrophic feature collapse, and show our PRO-DSC promotes to produce representations that are aligned with a union of orthogonal subspaces. Therefore, our PRO-DSC provides a general framework for self-expressive model based deep subspace clustering.
>
> - **Scalablity**: In the implementation of our PRO-DSC, rather than directly optimizing the variables $Z$ and $C$, we optimize the parameters which are used to reparameterize them. Such a reparameterization strategy makes our implimentation is scalable to large-scale dataset and enjoy a generalization ability to out-of-sample unseen data.
>
> - **Extensibility**: Note that our PRO-DSC with a vanilla self-expressive model has already demonstrated highly competitive performance. Both AGCSC and ARSSC improve the self-expressive model by incorporating GCN modules and self-adaptive attention mechanisms, respectively. Thus, it will be attempting to employ them into the deep self-expressive models under our PRO-DSC framework, and we believe that extending PRO-DSC with current SOTA self-expressive models (such as AGCSC and ARSSC) would be more promising.
>
> Thus, as a general framework for self-expressive model based deep subspace clustering, our PRO-DSC is reasonable, scalable and flexible to miscellaneous extensions.
>
> For clarity, we have also submitted a revised version of our paper. We hope that our point-to-point responses could address the reviewer's questions and concerns well, and we are very glad to answer any further questions.

---

> > ### Author Response · Authors · 2024-11-25
> > **Kindly invite you to check our responses**
> >
> > We would like to thank the reviewer for the time and effort in reviewing our paper. We have carefully addressed the comments and the concerns point-to-point in detail. We hope you might find our responses satisfactory. As the discussion phase is about to close, we are very much looking forward to hearing from you about any further feedback. We will be very happy to clarify further concerns (if any).
> >
> > Thank you for your time.
> >
> > Best regards,
> > Authors

---

> > > ### Comment · Reviewer_JbyQ · 2024-11-27
> > > **Thanks for the response**
> > >
> > > Dear authors,
> > >
> > > Thank you for your detailed response. It addressed most of my concerns, and I will consider raising my score.
> > >
> > > Best,
> > > Reviewer

---

> ### Author Response · Authors · 2024-11-27
>
> We would like to thank the reviewer for the valuable time and effort in reviewing our paper, and taking time to read our responses.  And we sincerely appreciate the reviewer for increasing the rating. Thank you very much!
>
> Best Regards,
>
> Authors

---

### Official Review · Reviewer_6RSz · 2024-10-30

**Soundness:** 3
**Presentation:** 3
**Contribution:** 3
**Rating:** 6
**Confidence:** 4

**Summary:**

This paper studies the deep subspace clustering problem. The general framework of the existing algorithms suffers from feature collapse and lacks a theoretical guarantee to learn the desired UoS representation. This paper presents a principled framework for deep subspace clustering (PRO-DSC), which is designed to learn structured representations and self-expressive coefficients in a unified manner. The motivation is clear, and the experimental performance of the proposed model is also good.

**Strengths:**

1.	The motivation is clear.
2.	The proposed method has strong theoretical support.
3.	The problem that needs to be solved is important, and the proposed method is reasonable.

**Weaknesses:**

In Eq. (4), it needs to be clarified why the logdet term is adopted. Is it only used to solve the representation collapse problem? Are there any other methods that can solve this collapse problem? More importantly, what are the underlying physical meanings of this term?

In Table 2, although the proposed methods seem to have the best performance. Some really SOTA deep clustering methods are not compared, like
[1] Learning Representation for Clustering Via Prototype Scattering and Positive Sampling, 2023 TPAMI.
[2] Towards Calibrated Deep Clustering Network, 2024 Arxiv.

Ablation studies should be performed to check the effectiveness of each component of the proposed method.

**Questions:**

See the weakness.

---

> ### Author Response · Authors · 2024-11-21
> **1. Regarding to using the $\log \det(\cdot)$ term**
>
> We highly appreciate the reviewer for recognizing our work as a reasonable method with clear motivation and strong theoretical support to address an important problem, for the supportive rating, and for the insightful and constructive comments.
>
> ### **1.1 Why is $\log \det(\cdot)$ introduced?**
>
> The reason to introduce the $\log \det(\cdot)$ term is to address the catastrophic feature collapse issue in the self-expressive model based deep subspace clustering (SEDSC), which is a longstanding issue to trap the development of deep subspace clustering method with self-expressive framework. Moreover, using the $\log \det(\cdot)$ term also brings two desirable properties.
>
> - **Promoting the diversity of the learned representations.**
>
>   In Theorem 2, we prove that the representation $Z$ learned by our PRO-DSC achieves the maximum rank, implying that the leaned features make the best use of the entire feature space, thereby preserving as much information as possible. The theoretical result is empirically verified (Please see the first column in Fig. B.3). As observed, the singular value of the representations learned by PRO-DSC is larger than CLIP's, implying that the features of our PRO-DSC are more diverse and uniform in the feature space.
>
> - **Promoting orthogonality among different subspaces.**
>
>   In Theorem 3, we show that PRO-DSC can promote orthogonality among different subspaces, which implies that representations of different clusters are well separated. The theoretical result is also supported empirically (Fig. 3 and Fig. B.2).
>
>   Notably, the $\log \det(\cdot)$ term plays indispensable role for both properties. For a strict analysis, we refer the reviewer to check the proofs of Theorem 2 and Theorem 3. Briefly, it serves as a core part of the objective function of the water-filling problem in the proof of Theorem 2, and it enables the convexity of the objective function can be used in the proof of Theorem 3.
>
> ### **1.2  Other methods to solve the collapse problem? What is and why logdet()? Physical meanings?**
>
> - **About other methods to solve the collapse problem?**
>
>   The catastrophic feature collapse problem implies that the learned representations occupy merely a tiny subspace. Intuitively, we believe that other methods which can increase the volume of the representation or the dimensionality of the subspace spanned by the representation could potentially solve the collapse issue.  For example, the rank of $\boldsymbol{Z}$, and the nuclear norm of $\boldsymbol{Z}$, i.e.,$\Vert \boldsymbol{Z}\Vert_* = \sum_{i=1}^N \sigma_{\boldsymbol{Z}}^{(i)}$, which is the sum of the singular values of the subspace spanned by the learned representations.
>
> - **Why is $\log \det(\cdot)$ used?**
>
>   The rank is the direct measure of dimensionality of the subspace of the learned features, but it is non-differentiable and difficult to optimize. In practice, there are two common surrogates for the rank: the nuclear norm, which is defined as $\Vert \boldsymbol{Z}\Vert_* := \sum_{i=1}^N \sigma_{\boldsymbol{Z}}^{(i)}$ and
>   $\frac{1}{2}\log\det(\boldsymbol{I}+\alpha \boldsymbol{Z}\boldsymbol{Z}^\top)=\frac{1}{2}\sum_{i=1}^N\log(1+\alpha (\sigma_{\boldsymbol{Z}}^{(i)})^2)$.
>   The reason to use $\log \det(\cdot)$, rather than the nuclear norm is that, the $\log \det(\cdot)$ is the differentiable surrogate of the rank, which is tighter than the nuclear norm (see Fazel et al 2003).
>
> - **Physical meanings of $\log \det(\cdot)$?**
>
>   In (Ma et al. TPAMI 2007), the meaning of $\log\det(\boldsymbol{I}+\alpha \boldsymbol{Z}\boldsymbol{Z}^\top)$ is explained using a sphere packing from the perspective of information theory. Suppose that the representations are distorted by the random noise, i.e., $\hat z_i = z_i +\epsilon_i$, where $\epsilon_i \sim \mathcal{N}(0,\frac{\varepsilon^2}{d}\boldsymbol{I})$, then the volume $\text{vol}(\boldsymbol{\hat Z})$ of the region spanned by the representation vectors $\{z_i\}$ and the volume of noise $\text{vol}(\boldsymbol{\epsilon})$ are defined respectively as follow:
> $$
> \begin{align}
>     \text{vol}(\boldsymbol{\hat Z}) & \propto \sqrt{\det(\hat \Sigma_\boldsymbol{Z})},\\ \\ \\ ~\hat \Sigma_\boldsymbol{Z} = \mathbb{E}_ \\boldsymbol{\epsilon} \left[\frac{1}{N}\sum_{i=1}^N \hat z_i z_i^\top\right] = \frac{\varepsilon^2}{d}\boldsymbol{I} + \frac{1}{N}\boldsymbol{Z}\boldsymbol{Z}^\top, \\\\
>     \text{vol}(\boldsymbol{\epsilon}) & \propto \sqrt{\det(\frac{\varepsilon^2}{d}\boldsymbol{I})}.
> \end{align}
> $$
>
> Therefore, we see that $\log \det(\cdot)$, as a measure of the size of the subspace spanned by the representations, approximates the coding length by the noisy sphere packing with binary coding, that is:
> $$
> \- \begin{align}
>       R(\boldsymbol{Z}) = \log_2(\text{\\# of spheres}) \\propto \log_2\frac{\text{vol}(\boldsymbol{\hat Z})}{\text{vol}(\boldsymbol{\epsilon})} = \frac{1}{2}\log_2{\det(\boldsymbol{I}+\frac{d}{N\varepsilon^2}\boldsymbol{Z}\boldsymbol{Z}^\top)}.
>   \end{align}
> $$

---

> ### Author Response · Authors · 2024-11-21
> **2. Regarding to Comparison to SOTA Deep Clustering Methods**
>
> We thank the reviewer for bringing two recent SOTA methods to our attention.
>
> The performance comparison to ProPos [1] and CDC [2] is shown below, where the results of ProPos and CDC are directly cited from the respective papers.
>
> | Method         | CIFAR10     | CIFAR20     |
> | -------------- | ----------- | ----------- |
> | ProPos         | 94.3 / 88.6 | 61.4 / 60.6 |
> | ProPos w/o PSL | 87.9 / 79.4 | 55.0 / 57.0 |
> | CDC            | 94.9 / 89.3 | 61.7 / 60.9 |
> | CDC w/o init   | 89.4 / 86.5 | 44.4 / 52.3 |
> | **PRO-DSC**        | 92.7 / 85.9 | 59.0 / 60.4 |
>
> Although ProPos and CDC achieve superior performance, their results primarily depend on introducing a novel Prototype Scattering Loss (PSL) and a specific initialization.  While the performance of our PRO-DSC is not the best, it is still quite competitive, because currently our PRO-DSC is a vinila self-expressive model based deep subspace clustering (SEDSC) framework without ``bells and whistles''.
>
> - PSL obtains the posterior probabilities of the data categories by performing $k$-means on the entire dataset at each epoch. During the training, the posterior probabilities are used to compute the prototypes of each category, upon which the prototype contrastive learning is conducted.
> - The initialization of CDC involves performing $k$-means on the entire dataset and using the cluster centers as the initialization for the fully connected layer. This initialization step is iterated until the entire MLP network is initialized.
>
> In summary, the leading performance of PSL and CDC over our PRO-DSC mainly is due to their effectively exploiting the supervision of the pseudo-label information; whereas our PRO-DSC framework is merely a deep version of the most basic self-expressive model without exploiting the pseudo-label information.
>
> Our PRO-DSC attempts to provide a reasonable deep subspace clustering framework with theoretical justification, especially to tackle the big challenge that traps the self-expressive model based deep subspace clustering (SEDSC) to learn the representations with desired structure (i.e., a union of subspaces).  We provided theoretical justification to demonstrate that our PRO-DSC can prevent the catastrophic feature collapse and the empirical results (See Figs.1 and 2) also confirm our theoretical findings. Moreover, we conducted extensive experiments on deep clustering benchmarks to evaluate the performance of our PRO-DSC. Currently, our framework is implemented naively without special designs or tricks for the leaderboard of the performance; therefore, the reported performance of our PRO-DSC may not be the absolutely best.
>
> As future work, we are interested in enhancing the performance of our PRO-DSC to achieve the best possible clustering results by incorporating the ideas from the works [1][2], and the work in the field of subspace clustering that uses pseudo-label information to achieve excellent results (e.g., Li et al., TIP 2017).
>
> [1] Huang et al., "Learning representation for clustering via prototype scattering and positive sampling", IEEE Transactions on Pattern Analysis and Machine Intelligence 45 (6), 2022: 7509-7524.
>
> [2] Jia et al., "Towards Calibrated Deep Clustering Network." arXiv preprint arXiv:2403.02998, 2024.
>
> [3] Li et al., "Structured sparse subspace clustering: A joint affinity learning and subspace clustering framework," IEEE Transactions on Image Processing 26 (6), 2017: 2988-3001.

---

> ### Author Response · Authors · 2024-11-21
> **3. Regarding to Ablation Studies**
>
> We kindly remind the reviewer that the ablation studies were included in the our previous submission in Table 4, where we evaluated the effectiveness of each component in our PRO-DSC framework and also evaluated the performance of using four different regularization term on the self-expressive coefficients. For any question, we are very happy to discuss them further.
>
> For clarity, we have also submitted a revised version of our submission. And we hope that our point-to-point responses could address the reviewer's questions well, and we are glad to answer any further question regarding to our submission.

---

> > ### Author Response · Authors · 2024-11-25
> > **Kindly invite you to check our point-to-point replies**
> >
> > We would like to thank the reviewer for the time and effort in reviewing our paper. We have carefully addressed the comments and the concerns in detail. We hope you might find our responses satisfactory. Since that the discussion phase is approaching to close, we are very much looking forward to hearing from you about any further feedback. We will be very happy to clarify further concerns (if any).
> >
> > Thank you for your time.
> >
> > Best regards,
> >
> > Authors

---

> ### Author Response · Authors · 2024-11-27
>
> We thank the reviewer for the valuable time and the effort in reviewing our paper.
>
> In the rebuttal, we have clarified the reason to adopt the $\log \det(\cdot)$ term, interpreted the other choices and the physical meanings, and discussed the performance comparison to the mentioned methods. And we have revised our paper accordingly (see L131-134, L1669-1676).
>
> We hope you might find our responses and revisions satisfactory, and sincerely hope you will reconsider your rating based on our clarification in responses and the revised paper. Thank you for your time!
>
> Best Regards,
>
> Authors

---

> ### Author Response · Authors · 2024-12-02
> **Kindly invite you to check our replies**
>
> We thank the reviewer for the valuable time and the effort in reviewing our paper. Since that the discussion phase is approaching to close, we are very much looking forward to hearing from you about any further feedback. We hope you might find our responses and revisions satisfactory, and sincerely hope you will reconsider your rating based on our clarification and the revised paper.
>
> Thank you for your time!
>
> Best Regards,
>
> Authors

---

### Official Review · Reviewer_eucv · 2024-11-03

**Soundness:** 3
**Presentation:** 3
**Contribution:** 2
**Rating:** 5
**Confidence:** 4

**Summary:**

This paper proposed a Principled fRamewOrk for Deep Subspace Clustering (PRO-DSC), which is designed to learn structured representations and self-expressive coefficients in a unified manner. First, PRO-DSC incorporates an effective regularization into self-expressive model to prevent the catastrophic representation collapse with theoretical justification. Second, PRO-DSC demonstrated that it is able to learn structured representations that form a desirable UoS structure, and also developed an efficient implementation based on reparameterization and differential programming. Comprehensive experiments verify the superiority of the proposed model.

**Strengths:**

1.The paper is well-written and technically sound.
2.The experiments are comprehensive.

**Weaknesses:**

1.What are the limitations and failed cases of the proposed method? Some discussion needed.
2.There may be insufficient implementation details provided, hindering reproducibility of the study and making it challenging for other researchers to replicate the results. Such as, the results for PRO-DSC are averaged over three trials (with±std), what about other methods? Their results are the best or mean? As I know, some methods like k-means and SC are sensitive to the initialization, their results are recorded by the best or mean in some runs with different initializations? A fair experimental setting is necessary.
3.The subspace description coefficients and manifold parts provided in the article are based on existing research results and lack sufficient innovation.

**Questions:**

1.What are the limitations and failed cases of the proposed method? Some discussion needed.
There may be insufficient implementation details provided, hindering reproducibility of the study and making it challenging for other researchers to replicate the results. Such as, the results for PRO-DSC are averaged over three trials (with±std), what about other methods? Their results are the best or mean? As I know, some methods like k-means and SC are sensitive to the initialization, their results are recorded by the best or mean in some runs with different initializations? A fair experimental setting is necessary.

---

> ### Author Response · Authors · 2024-11-21
> **1. Regarding to Limitations and Failed Cases**
>
> We thank the reviewer for recognition of our paper writing, techniques, and experiments.
>
> **Limitations:**  In this paper, we explore an effective framework for deep subspace clustering with theoretical justification. However, it is not clear how to develop the geometric guarantee for our PRO-DSC framework to yield subspace-preserving (correct) solution. Moreover, it is an unsupervised learning framework, we left the extension to semi-supervised setting as future work.
>
> **Failed cases:**  In this paper, we evaluate our PRO-DSC framework on two scenarios of synthetic data (Fig. 5), six benchmark datasets with CLIP features (Table 1), five benchmark datasets which are for training from scratch (Table 2), three out-of-domain datasets (Table B.9), using four different regularization terms (Table 4), using different feature extractor (Table B.8) and varing hyper-parameters (Fig. 7 and Table B.10). During the rebuttal, to reply Reviewer JbyQ, we also conducted experiments on two face image datasets (Extended Yale b and ORL), an object recognition (COIL-100), a text dataset (REUTERS-10k) and a time-seris dataset (UCI HAR). Currently, we did not find significant failure cases.
>
> However, as demonstrated in Fig. 1, our PRO-DSC will fail if the sufficient condition to prevent catastrophic feature collapse is not satisfied by using improper hyper-parameters $\gamma$ and $\alpha$. We have added the discussions on limitations and the failure cases in the revised version in the Appendix C.

---

> ### Author Response · Authors · 2024-11-21
> **2. Reproducibility and Experimental Details**
>
> To ensure the reproducibility of our work, we have submitted the anonymized source code. All datasets used in our experiments are publicly available, and we provided a comprehensive description of the data processing steps in Appendix B.1.
> Additionally, we also provided detailed experimental settings and configurations in Appendix B.1 to facilitate the reproduction of our results. We are very glad to answer any question about the details of reproducing our experimental results.

---

> ### Author Response · Authors · 2024-11-21
> **3. Regarding to Fairness of Reported Results**
>
> In Appendix B.1 of our submitted paper, we have provided the running details of EnSC, SSCOMP, DSCNet, and CPP.
>
> - For SENet, SCAN and EDESC, we adjust the hyper-parameters and repeat experiments for three times, with only the best results are reported.
>
> - For TEMI, we directly cited the results from the paper.
>
> - For our PRO-DSC, we repeated experiments for three trials and report the average results (with standard deviation).
>
> Nonetheless,  the two classical methods (i.e., k-means and spectral clustering) are overlooked to report the running details.
>
> - For k-means and spectral clustering (including when spectral clustering is used as the final step in subspace clustering), we repeat the clustering 10 times with different random initializations (by setting n$\\_$init=10 in scikit-learn) and report the best results.
>
> Moreover, during the rebuttal, we conducted more experiments on both the CLIP feature and the experiments of training from scratch to report the average results with standard deviation over 10 trials, and these results have been updated in the revised paper (See the updated results in Tables 1 and 2 which are highlighted in blue). Here, we also listed the updated results (ACC% / NMI%) in the following tables for reference.
>
> - **Updated experimental results on the CLIP feature in Table 1**
> | Datasets             | CIFAR10             | CIFAR100            | CIFAR20             | TinyImageNet        | ImgNetDogs-15       | ImgNet-1k           |
> | -------------------- | ------------------- | ------------------- | ------------------- | ------------------- | ------------------- | ------------------- |
> | **repeat 3 trials**  | 97.2±0.2 / 93.2±0.2 | 77.2±0.4 /82.4±0.3  | 71.4±1.3 / 73.3±0.4 | 69.4±1.4 / 80.4±1.1 | 83.6±0.1 / 81.5±0.4 | 65.1±0.3 / 83.6±0.2 |
> | **repeat 10 trials** | 97.2±0.2 / 92.8±0.4 | 77.3±1.0 / 82.4±0.5 | 71.6±1.2 / 73.2±0.5 | 69.8±1.1 / 80.5±0.7 | 84.0±0.6 / 81.2±0.8 | 65.0±1.2 / 83.4±0.6 |
>
> - **Updated experimental results of training from scratch in Table 2**
> | Datasets             | CIFAR10           | CIFAR100          | CIFAR20           | TinyImageNet      | ImgNetDogs-15     |
> | -------------------- | ----------------- | ----------------- | ----------------- | ----------------- | ----------------- |
> | **repeat 3 trials**  | 92.7±0.1 / 85.9±0.2 | 59.0±0.2 / 60.4±0.1 | 56.2±0.6 / 67.0±0.2 | 31.2±0.2 / 47.0±0.2 | 74.6±0.2 / 70.2±0.1 |
> | **repeat 10 trials** | 93.0±0.6 / 86.5±0.2 | 58.3±0.9 / 60.1±0.6 | 56.3±0.6 / 66.7±1.0 | 31.1±0.3 / 46.0±1.0 | 74.1±0.5 / 69.5±0.6 |

---

> ### Author Response · Authors · 2024-11-21
> **4. Regarding to the Novelty**
>
> While each term of the losses in our PRO-DSC framework has existed in prior literature, we argue that it is still for the first time that the term $-\log \det (\cdot)$ is introduced into the self-expressive loss to address the catastrophic collapse issue. Moreover, we have provided theoretical justification and extensive empirical evaluations to demonstrate that adding such a regularization term can indeed resolve the catastrophic feature collapse, which is a longstanding issue in the self-expressive model based deep subspace clustering (SEDSC). Besides, the efficient implementation enable our proposed PRO-DSC can handle large-scale, complex real-world dataset. We believe that our PRO-DSC can benefit the people who are working in subspace clustering or stuck in deep subspace clustering.
>
> For clarity, we have also submitted a revised version of our paper. And we hope that our point-to-point responses could address the reviewer's questions well, and we are very glad to answer any further question regarding to our submission.

---

> > ### Author Response · Authors · 2024-11-25
> > **Kindly invite you to check our point-to-point responses**
> >
> > We would like to thank the reviewer for the time and effort in reviewing our paper. We have carefully clarified the comments and the concerns in detail, and hope you might find our responses satisfactory. Since that the discussion phase is approaching to close, we are very much looking forward to hearing from you about any further feedback. We will be very happy to clarify further concerns (if any). Thank you for your time.
> >
> > Best regards,
> >
> > Authors

---

> ### Author Response · Authors · 2024-11-27
>
> We thank the reviewer for the valuable time and effort in reviewing our paper.
>
> In the rebuttal, we have added the discussions on the limitations and failure cases in Appendix C. Moreover, we have clarified the fairness in our experiments and the reproducibility of our work, and also clarified the significance of our work. In addition, we note that in the initial submission, we have submitted the code to reproduce the results in our paper.
>
> We hope you might find our responses and revisions satisfactory, and sincerely hope you will reconsider your rating based on our clarification in responses and the revised paper. Thank you for your time!
>
> Best Regards,
>
> Authors

---

> ### Author Response · Authors · 2024-11-28
>
> Dear Reviewer,
>
> Thank you for your time in reviewing our paper. We have updated our paper by adding discussions on limitations and failure cases in Appendix C and supplying more experimental results averaged on 10 trails in Tables 1 and 2. We hope you might find our responses and revisions satisfactory, and we sincerely hope you will reconsider your rating based on our responses and the updated paper. Thank you for your time and effort!
>
> Best Regards,
>
> Authors

---

> ### Author Response · Authors · 2024-12-02
> **Kindly invite you to check our replies**
>
> We thank the reviewer for the valuable time and the effort in reviewing our paper. Since that the discussion phase is approaching to close, we are very much looking forward to hearing from you about any further feedback.  We hope you might find our responses and revisions satisfactory, and sincerely hope you will reconsider your rating based on our clarification and the revised paper. Thank you for your time!
>
> Best Regards,
>
> Authors

---

### Official Review · Reviewer_KBNV · 2024-11-05

**Soundness:** 3
**Presentation:** 3
**Contribution:** 2
**Rating:** 8
**Confidence:** 5

**Summary:**

The paper studied deep subspace clustering. Existing deep subspace clustering methods suffer from feature collapse, where learned representations collapse into subspaces with dimensions much lower than the ambient space. The paper proposes to add a loss term that alleviates this issue, which is backed up with some theoretical study and experiments on real-world dataset.

**Strengths:**

1. Experiments on synthetic and real-world datasets are comprehensive, and the reviewer appreciate that.  Synthetic experiments: what happens when you add an additional subspace? Case 1: the subspace is 0-dimensional (points centered around a centroid) Case 2: you add a another curve as you have around the great circle, but now put it vertically. The subspaces will be intersecting. I am not expecting the methods to outperform anything, this is just for understanding the method better.

**Weaknesses:**

1. The reviewer is concerned with the novelty of the paper. The main motivation of the paper is the observation from Haeffele el al. 2021 that if one learns a representation and apply a subspace clustering type of loss on the representations, then the representations tend to collapse. Therefore, the paper proposes to incorporate an additional term (equation 3) into the loss to prevent collapse. This theme of combining subspace clustering loss and (equation 3) has been explored before: In Ding et al 2023, they used a combination of (equation 3) and the subspace clustering loss in Ma et al 2007. One could go ahead and try many different subspace clustering loss functions, but the contribution seems incremental apriori. If one reads the introduction of that paper, it appears that the motivation was rather similar to this one, but no discussion was given in the current paper.
2. The reviewer is also concerned with the theoretical contributions.
    1. Lemma 1 and its proof is not a contribution, as the paper clearly states they are from Haeffele et al. 2021.
    2. It is difficult to connect Theorem 1 with the main objective (equation 5). In particular, it is unclear at the optimality of (equation 5), whether (and why) the optimal Lagrangian multiplier nu satisfies the conditions in Theorem 1.
    3. Theorem 3: I do not quite understand the statement.
        1. First, apriori there might be multiple solutions to PRO-DSC. When you say ‘the’ optimal solution, what do you mean? Do you mean you have a sufficient condition, such that there exists ‘one’ optimal solution such that some ideal properties hold on this solution? Or do you mean ‘all’ optimal solutions
        2. Second, I am a bit confused on Z, C vs Z^*, C^*. Gamma is defined to permute (or ‘align’) columns of Z, but on line 238 they are used to permute Z^*. Is Gamma a function of Z or Z^*? In the sufficient condition &lt;(I-C)(I-C)^T, G-G^*>, should it be C^* instead? Or even a step back: how is C defined in Theorem 3?
        3. It is unclear what how to interpret sufficient condition &lt;(I-C)(I-C)^T, G-G^*>, e.g., how does it connect with Z^* lying in a union of subspaces or C^* being correctly connecting points only from the same subspace. It might strengthen the result a bit if there is a simple case (e.g., the subspaces are independent or orthogonal, points being uniformly spread within each subspace) where such conditions hold.

I am in general happy to adjust my ratings based on whether the rebuttal addresses the comments.

**Questions:**

See above

---

> ### Author Response · Authors · 2024-11-21
> **1. Regarding to the Novel Contributions**
>
> We appreciate the reviewer for reading our submission carefully and, for providing some critical yet valuable comments on the novelty of our work. However, we do not agree with the reviewer that our work is incremental.
>
> In the past decade, subspace clustering has been substantially put forward owning to:
> - a) self-expressive (SE) model ;
> - b) correctness guarantees for SE coefficients to satisfy a subspace-preserving property with respect to different regularization terms, and
> - c) scalable algorithms to solve SE model.  Though there are many heuristic methods to address subspace clustering problem, people usually attribute the substantial advancements in subspace clustering to clear algorithms with solid correctness (theoretical) guarantee.
>
> In the era of deep learning, developing a reasonable deep framework for subspace clustering has also attracted a lot of research attention in the past a few years. One of the most popular and elegant frameworks for deep subspace clustering is DSCNet (Ji et al., NeurIPS'17),  which encapsulates a SE model (which is called a SE layer) in the "middle" of a stacked convolutional auto-encoder network (SCAE) and is trained by using a combination of reconstruction loss, SE loss and $\ell_1$ or $\ell_2$ regularizer. Although remarkable clustering accuracy has been reported on four image datasets (especially on Extended Yale B), there is no clear evidence to demonstrate that SCAE in DSCNet is able to amend the input data to align with a union of subspaces (UoS). More critically, as theoretically revealed by (Haeffele et al. ICLR'21), the optimal solution of DSCNet (which is called SE based Deep Subspace Clustering, SEDSC) suffers from a catastrophic feature collapse---which thus leads to a seriously degenerated performance. Unfortunately, Haeffele et al. (ICLR'21) did not provide a reasonable framework for deep subspace clustering.
>
> An interesting recent work is MLC (Ding et al. ICCV'23), which is inspired by MCR2 (Yu et al. NeurIPS'20). In MLC, the loss function is modified from MCR2, by changing the membership matrix in MCR2 to a doubly stochastic affinity matrix, and adding a (negative) entropy regularization on the affinity. However, there is no theoretical justification on the optimal solution and the algorithm is sensitive to the initialization.
>
> The goal of our work is to provide a reasonable framework for deep subspace clustering. To keep clarity and simplicity, we devote a principled solution to make SEDSC to avoid catastrophic feature collapse and to learn representation with desired structure, e.g., a union of subspaces.  We argue that:
> - a) incorporating the total coding rate term as defined in Eq. (3) into SEDSC model as a regularizer has not existed in the literature, and thus it is novel; and
> - b) providing theoretical justification to verify the effect of incorporating such a regularizer to prevent the catastrophic feature collapse is novel contribution.
>
> Incorporating the total coding rate term into SEDSC is natural and elegant, and it has NOT been investigated in the literature yet. Moreover, it is NOT an arbitrary "combining subspace clustering loss and Eq. (3)". We don't think that there are "many different subspace clustering loss functions" can be easily combined and supported by solid theoretical justification. Actually, for our PRO-DSC, as a general framework, we have investigated at least four different types of regularizers on the self-expressive coefficients (See Ablation Studies in Table 4).
>
> In the revised paper, we have added a discussion to connect and compare MLC (Ding et al. ICCV '23) in the end of Related Work.  Thanks the reviewer for the careful reading and the suggestions. For any further question, we are very glad to discuss and reply.

---

> ### Author Response · Authors · 2024-11-21
> **2. Regarding to Lemma 1 and its Proof**
>
> We note that Lemma 1 had clearly marked where it comes from. The reason to provide the proof for Lemma 1 in Appendix A is to serve as the background prerequisite and the preparatory of the following proofs.

---

> ### Author Response · Authors · 2024-11-21
> **3. Regarding to Theorem 1**
>
> Thank the reviewer for careful checking our informal statement of Theorem 1. The formal and complete statement of Theorem 1 is provided in Appendix A.
>
> Note that $\nu$ is a Lagrangian multiplier (which is also called a dual variable in its Lagrangian dual problem). We add a paragraph of justification (which is highlighted in blue in the Appendix A. of the revised version) for the existence of the optimal Lagrangian multiplier $\nu_\star$ in Appendix A.
>
> Our analysis is based on the KKT optimality condition of problem in Eq. (5) and reveals that $\nu_\star \le \frac{\alpha}{1+\alpha}$ or $\nu_\star \le \frac{\alpha}{\frac{d}{N}+\alpha}$ in the case of $d\ge N$ and $d<N$. Thus, when the inequality in each case $\frac{\alpha}{1+\alpha}< \alpha - \gamma \lambda_{max}(\boldsymbol{M})$ or $\frac{\alpha}{\frac{d}{N}+\alpha}< \alpha - \gamma \lambda_{max}(\boldsymbol{M})$ holds, i.e., $\gamma \lambda_{max}(\boldsymbol{M})  < \frac{\alpha^2}{1+\alpha}$ or $\gamma \lambda_{max}(\boldsymbol{M})  < \frac{\alpha^2}{\frac{d}{N}+\alpha}$, then the condition of Theorem 1 will be automatically satisfied. This result implies that, we are only required to adjust the hyper-parameters $\gamma$ and $\alpha$ to satisfy the inequality $\gamma \lambda_{max}(\boldsymbol{M})  < \frac{\alpha^2}{1+\alpha}$ or $\gamma\lambda_{max}(\boldsymbol{M})<\frac{\alpha^2}{\frac{d}{N}+\alpha}$, no need to concern about the specific value of $\nu_\star$.
>
> The justification is added in the Appendix A of the revised version.

---

> ### Author Response · Authors · 2024-11-21
> **4. Regarding to Theorem 3**
>
> ### **1)  About the Optimal Solutions**
>
> In this context, we are referring to 'all' optimal solutions. In the proof of Theorem 3, the inequalities are based on the property of convex function, where the equality holds for all the optimization variables $\boldsymbol{Z}$ and $\boldsymbol{C}$ which satisfy that both $\boldsymbol{G}$ and $\boldsymbol{C}$ are block diagonal under certain permutation $\Gamma$.  Thus, the conclusion in Theorem 3 does not depend on a specific global optimal solution.
>
> We note that the optimal solutions of PRO-DSC is NOT unique, since that a multiplication of an orthogonal matrix to $\boldsymbol{Z}$ does not change the objective value and constraint (see the results on synthetic data in Fig. 5). This means that although PRO-DSC is a nonconvex problem, it possesses an elegant rotation symmetry. We conjecture that all the local minimizers up to rotation symmetry are equivalently good [1], and we left the complete analysis of the optimization landscape of our PRO-DSC as a future work. We highly appreciate the reviewer for the insightful questions which remind us to notice of the optimization landscape. That will be left for feature work.
>
> [1] J. Wright and Y. Ma, High-Dimensional Data Analysis with Low-Dimensional Models: Principles, Computation and Applications, 2022. (Chapter 7)
>
> ### **2) About $\Gamma$ and the CSC condition in Theorem 3**
>
> We apology for the confusion in using the notion of permutation matrix $\Gamma$ in Theorem 3.  Note that multiplying a permutation matrix on the right (or left) rearrange the columns (or rows) of the matrix. The purpose of rearrangement is to group together the data points that potentially belong to the same subspace into the same block, which allows us to observe the structures of the learned representations and the self-expressive matrix. For simplicity and without loss of generality, we assume that the columns of $\boldsymbol{Z}$ are arranged into blocks with a certain permutation matrix $\boldsymbol{\Gamma}$, i.e., $\boldsymbol{Z} =[\boldsymbol{Z}_1,\boldsymbol{Z}_2,\cdots, \boldsymbol{Z}_k]$. Thus, the contents of Theorem 3 have also been carefully revised. Please refer to the revised version of our submission. Thanks the reviewer for pointing out this confusion issue about abuse of $\boldsymbol{\Gamma}$.
>
> ### **3)  About the CSC Condition and Interpretations**
>
> We consider the solution to satisfy the CSC condition as a (non-empty) feasible set, then the optimal $(\boldsymbol{Z}_ \\star,\boldsymbol{C}_ \\star)$ will make $(\boldsymbol{G}_ \\star, \boldsymbol{C}_ \\star)$ be block-diagonal with respect to certain permutation $\boldsymbol{\Gamma}$.
>
> On the other hand, for a given $(\boldsymbol{Z},\boldsymbol{C})$ (at a certain iteration or when converged), suppose that we arrange $\boldsymbol{Z}$ with respect to the ground-truth labels, then the CSC condition can serve as a computable metric of the distance about $(\boldsymbol{Z},\boldsymbol{C})$ to the desired arrangement of the orthogonal subspaces and $\boldsymbol{Z}_ \\star$ and the (correct) block diagonal coefficient matrix $\boldsymbol{C}_ \\star$ (as in Figure 4).
>
> For a given $(\boldsymbol{Z},\boldsymbol{C})$,  we can provide three simple examples to interpret CSC:
>
> - When the subspaces of $\boldsymbol{Z}$ are orthogonal, since $\boldsymbol{G}-\boldsymbol{G}^*=0$, the condition is automatically satisfied.
>
> - When $\boldsymbol{C}$ is a block diagonal matrix, the condition is also satisfied because $\boldsymbol{G}-\boldsymbol{G}^*$ is an off-block diagonal matrix.
>
> - When the subspaces associated with $\boldsymbol{Z}_1, \boldsymbol{Z}_2,\cdots,\boldsymbol{Z}_k$ are independent, since that self-expressive model yields block diagonal $\boldsymbol{C}$ for any regularizer that satisfies the Enforced Block Diagonal (EBD) conditions (see Lu et al 2018) when $\boldsymbol{Z}$ is fixed, CSC is also satisfied.
>
> Additionally, in Figure 4, our experimental results show that, even though there are fluctuations early in the optimization process, the CSC condition gradually holds as the training progresses.

---

> ### Author Response · Authors · 2024-11-21
> **5. More results on synthetic data**
>
> The synthetic experiments of adding an additional subspace are presented in Figure B.1 of Appendix B.3 in the revised version.
>
> - In case 1, we implement two sets with 100 points in each cluster sampled from Gaussian distribution $\boldsymbol{x}\sim\mathcal{N}([\frac{1}{\sqrt 2},0,\sqrt 2]^\top,0.05\boldsymbol{I}_3)$ and $\boldsymbol{x}\sim\mathcal{N}([-\frac{1}{\sqrt 2},0,\sqrt 2]^\top,0.05\boldsymbol{I}_3)$ in the same side of the sphere. PRO-DSC eliminates the nonlinearity in representations and maximally separates the different subspaces.
>
> - In case 2, we add a vertical curve with 100 points sampled by:
> $$
> \begin{equation}
>     \boldsymbol{x}=\begin{bmatrix}
>         \cos\left(\frac{1}{5}\sin\left(5\varphi\right)\right)\cos\varphi\\\\
>         \sin\left(\frac{1}{5}\cos\left(5\varphi\right)\right)\\\\
>         \cos\left(\frac{1}{5}\sin\left(5\varphi\right)\right)\sin\varphi
>     \end{bmatrix}+\boldsymbol{\epsilon},
>  \end{equation}
> $$
>  where $\boldsymbol{\epsilon}\sim\mathcal{N}(\mathbf{0},0.05\boldsymbol{I}_3)$ and use $\sin(\frac{1}{5}\cos(5\varphi))$ to avoid overlap in the intersection of the two curves. For the experiments on synthetic data, the learnable mappings $\boldsymbol{h}(\cdot;\boldsymbol{\Psi})$ and $\boldsymbol{f}(\cdot;\boldsymbol{\Theta})$ are implemented with two MLPs with Rectified Linear Units (ReLU) as the activation function. The hidden dimension and output dimension of the MLP is set to $100$ and $3$, respectively.
> We observed that PRO-DSC finds difficulties to learn desired representations for a subset of data points lying at and nearby the intersection of subspaces. However, those data points away from the intersection are linearized well.
>
> The detailed settings for our PRO-DSC are given as follows.
> - In case 1, we train PRO-DSC with batch-size $n_b=300$, learning rate $\eta=5\times 10^{-3}$ for $5000$ epochs and set $\gamma=1.3,\beta=500,\alpha=3/0.1\cdot 300$.
> - In case 2, we train PRO-DSC with batch-size $n_b=200$, learning rate $\eta=5\times 10^{-3}$ for $8000$ epochs and set $\gamma=0.5,\beta=500,\alpha=3/0.1\cdot 200$.
>
> For clarity, we submitted a revised version of our paper. And we hope that our point-to-point responses could address the reviewer's questions well, and we are very glad to answer any further question.

---

> > ### Author Response · Authors · 2024-11-25
> > **Kindly invite you to check our point-to-point responses**
> >
> > We would like to thank the reviewer for the time and effort in reviewing our paper. We have carefully addressed the comments in detail, and hope you might find our responses satisfactory. Since that the discussion phase is approaching to close, we are very much looking forward to hearing from you about any further feedback. We will be very happy to clarify further concerns (if any). Thank you for your time.
> >
> > Best regards,
> >
> > Authors

---

> ### Author Response · Authors · 2024-11-27
>
> We appreciate the reviewer for the valuable time and effort in reviewing our paper.
>
> We have clarified the significance of our work, justified the existence of the optimal dual variable, reorganized Theorem 3, added extra experiments, and revised our paper accordingly.
>
> We hope you might find our responses and revisions satisfactory, and sincerely hope you will reconsider your rating based on our clarification in responses and the revised paper. Thank you for your time.
>
> Best Regards,
>
> Authors

---

> ### Author Response · Authors · 2024-12-02
> **Kindly invite you to check our replies**
>
> We thank the reviewer for the valuable time and the effort in reviewing our paper. Since that the discussion phase is approaching to close, we are very much looking forward to hearing from you about any further feedback.  We hope you might find our responses and revisions satisfactory, and sincerely hope you will reconsider your rating based on our clarification and the revised paper. Thank you for your time!
>
> Best Regards,
>
> Authors

---

> > ### Comment · Reviewer_KBNV · 2024-12-03
> >
> > Thanks to the authors for their detailed response! I am happy to see that the theoretical results are much clearer than before and  I have thus increased my rating accordingly.

---

> ### Author Response · Authors · 2024-12-03
>
> We are very glad that the reviewer has satisfied our responses and revisions, and we sincerely appreciate the reviewer for the supportive rating. The constructive comments have really helped us to improve the quality of our paper. Thank you very much!

---

### Author Response · Authors · 2024-11-26
**Summary of Our Revisions**

We would like to thank all reviewers for the time and effort in reviewing our paper.

To address reviewers' comments and concerns, we have made the following changes.

- We have clarified the existence of the optimal dual variable $\nu_\star$ in Theorem 1 by deriving its upper bound, and provided a detailed theoretical justification in **Appendix A**.  (L1036-1087) (**KBNV**)

- We have revised the contents of Theorem 3 by properly giving a permutation matrix $\Gamma$ and using a relaxed CSC condition, and slightly modified the proof in **Appendix A**.  (**KBNV**)

- We have clarified the detailed settings for the reported experimental results and updated the results in Table 1 and Table 2 over 10 trials. (**eucv** and **JbyQ**)

- We have conducted experiments on datasets Reuters-10k and UCI HAR and added experimental results in Table B.4 of **Appendix B**. (**JbyQ**)

- We have conducted experiments to compare with AGSSC and three versions of ARSSCs and added the experimental results in Table B.5 of **Appendix B**.  (**JbyQ**)

- We have added the discussions on limitations and failure cases in **Appendix C**. (See L1654-1668) (**eucv**)

- We have add notes on using $\log \det (\cdot)$ and discussions on extensibility of our PRO-DSC in **Appendix C** (L131-134; L1669-1676).  (**6RSz** and **JbyQ**)

The revised contents are highlighted **in blue** in the revised paper. **We kindly invite all reviewers to take a look at these updated results and our responses.**

We sincerely thank all reviewers again for their valuable suggestions, which have greatly helped improve the quality of our work. If you have any further questions, we would be very happy to discuss them further.

---

### Meta-Review · Area_Chair_TWTH · 2024-12-19

**Metareview:**

This paper presents a principled framework for deep subspace clustering (PRO-DSC), which is designed to learn structured representations and self-expressive coefficients in a unified manner. First, PRO-DSC incorporates an effective regularization to prevent catastrophic representation collapse. Second, PRO-DSC learns structured representations that form a desirable UoS structure. Both theoretical and experimental analyses demonstrate the effectiveness of the proposed method.

However, I feel that some contributions might be overclaimed. First, the hyper-parameter configuration of PRO-DSC is elaborately tuned on different datasets, which plays an important role in its performance. Second, though the authors claim PRO-DSC could be trained from scratch, it still relies on a powerful representation learning baseline BYOL, which is also elaborately tuned for different datasets.

Based on the reviewers' comments, I decided to accept this paper. Nevertheless, the authors must carefully check the claims about the superior performance and ability to "train from scratch." Also, the potential limitation in the robustness of PRO-DSC should be explicitly discussed.

**Additional Comments On Reviewer Discussion:**

In general, the authors have addressed the reviewers' major concerns about the novelty, theoretical analyses, and experimental comparisons of this work. Two reviewers did not respond to the authors, while the other two reviewers were satisfied with the author's response and raised their scores accordingly.

---

### Decision · Program_Chairs · 2025-01-22

Accept (Poster)